

# Modeling dust sources, transport, and radiative effects at different altitudes over the Tibetan Plateau

[1]Zhiyuan Hu, [1]Jianping Huang, [2]Chun Zhao, [3]Qinjian Jin, [4]Yuanyuan Ma, [5]Ben Yang

5    [1]Key Laboratory for Semi-Arid Climate Change of the Ministry of Education, College of Atmospheric Sciences, Lanzhou University, Lanzhou 730000, China

[2]School of Earth and Space Sciences, University of Science and Technology of China, Hefei, Anhui, China.

[3]Department of Earth and Atmospheric Sciences, Cornell University, Ithaca, New York 14853, USA

10   [4]Key Laboratory of Land Surface Process and Climate Change in Cold and Arid Regions, Northwest Institute of Eco-Environment and Resources, Chinese Academy of Science, Lanzhou 730000, China

[5]School of Atmospheric Sciences, Nanjing University, Nanjing, China

*Corresponding author: Jianping Huang

phone: 0931-8914282

email: hjp@lzu.edu.cn



**Abstract**

Mineral dust plays an important role in the climate of the Tibetan Plateau (TP) by modifying the radiation budget, cloud macro- and microphysics, precipitation, and snow albedo. Meanwhile, the TP with the highest topography in the word can affect intercontinental transport of dust plumes and induce typical distribution characteristics of dust at different altitudes. In this study, we conduct a quasi-global simulation to investigate the characteristics of dust source contribution and transport over the TP at different altitude by using a fully coupled meteorology-chemistry model (WRF-Chem) with a tracer-tagging technique. Generally, the simulation reasonably captures the spatial distribution of satellite retrieved dust aerosol optical depth (AOD) at different altitudes. Model results show that dust particles are emitted into atmosphere through updrafts over major desert regions, and then transported to the TP. The East Asian dust is transported and lifted up to the TP, contributing a mass loading of 50 mg/m$^2$ at 3 km height and 5 mg/m$^2$ at 12 km height over the northern slop of the TP. Dust from North Africa and Middle East are concentrated over both northern and southern slopes below 6 km, where mass loadings range from 10 to 100mg/m$^2$ and 1 to 10 mg/m$^2$ below 3 km and above 9 km, respectively. As the dust is transported to the north and over the TP, mass loadings are 5-10 mg/m$^2$ above 6 km.

The imported dust mass flux from East Asia to the TP is 7.9 Tg/year mostly occuring at the heights of 3–6 km. The North African and Middle East dust particles are transported eastward following the westerly jet, and then imported into the TP at West side with the dust mass flux of 7.8 and 26.6 Tg/year, respectively. The maximum mass flux of the North African dust mainly occurs in 0–3 km (3.9 Tg/year), while the Middle East within 6–9 km (12.3 Tg/year). The dust outflow occurs at East side (−17.89 Tg/year) and South side (−11.22 Tg/year) of the TP with a peak value (8.7 Tg/year) in 6–9 km . Moreover, the dust mass is within the size range of 1.25~5.0 $\mu$m and dust number is concentrated in the size range of 0.156~1.25 $\mu$m. Compared with other aerosols, the dust contributes more than 50% to the total AOD over the TP. The direct radiative forcing induced by the dust is −1.28 W/m$^2$ at the top of the atmosphere (cooling), 0.41 W/m$^2$ in the atmosphere (warming) and −1.68 W/m$^2$ at the surface (cooling). Our quantitative analyses of the dust contribution from different source regions and the associated radiative forcing can help better understand the role of dust on the climate over the TP and surrounding regions.





## 1 Introduction

Mineral dust, emitted from the deserts such as Sahara, Arabian, Gobi and Taklamakan, is a major contributor to the global aerosol burden (e.g., Yu et al., 2012; Hu et al., 2016; Uno et al., 2009). Dust aerosols can be lifted into upper-troposphere by the strong frontal/postfrontal convection from major desert source regions, and then transported to downwind regions following westerlies (Zhao et al., 2006; Hu et al., 2019), with parts of the dust even being transported for one full circuit around the globe (Uno

et al., 2009). During the intercontinental transport, dust has a significant effect on the regional and global climate system (Lau et al., 2006; Jin et al., 2015, Pu et al., 2018). For instance, previous studies showed that dust could change the atmospheric heating and affect atmospheric circulation patterns in the mid- and upper-troposphere (Lau et al. 2006; Lau and Kim 2006; Alizadeh-Choobari et al., 2014). Further, dust could spur anomalous water cycle feedback and alter the rainfall distribution over East Asia (Lau et

al., 2016; Liu et al., 2011) and South Asia (Lau and Kim 2006; Kim et al., 2017; Jin et al., 2014, 2017). Dust particles were also found to change regional radiation budget (Huang et al., 2005, 2006; Zhao et al., 2014) by scattering and absorbing the solar radiation, and modify the properties of cloud (Li et al., 2017, 2018) by serving as cloud condensation nuclei and ice nuclei. Dust aerosols also exacerbate air quality (Chin et al., 2007; Hu et al., 2016) and reduce regional visibility (Watson, 2002). When deposited on

snow/ice surface, dust could accelerate the snowmelt and glacier retreat, and change the hydrological cycle (Qian et al., 2009; Painter et al., 2010; Huang et al., 2011; Zhao et al., 2014); when deposited into rainforests and ocean, dust could provide nutrients to ecosystems and influence carbon cycle (Mahowald et al., 2008; Yu et al., 2015).

    As the highest (about 4000 m) and largest plateau in the world, the Tibetan Plateau (TP) has received

extensive scientific attentions in the past decades due to its unique topographic characteristics (Ma et al., 2014; Yao et al., 2012). Frequently, the TP is attacked by the intercontinental transport dust (Huang et al., 2008; Liu et al., 2008) from surrounding major deserts, i.e., Gobi and Taklamakan deserts on the northern side and Sahara and Arabian deserts on the western side, and the regional climate over the TP has been greatly influenced (Lau et al., 2006; Wu et al., 2007). Meanwhile, the TP plays an important role in the

dust global-scale transport through dynamical and thermal forcing in the mid-latitude (Xu et al., 2018). When dust plumes arrive at the TP, they are split into two branches and stack up around the slopes of the TP (Lau et al., 2006; Qin et al., 2016; Hu et al., 2019); then dust particles climb to the TP due to the



topographic uplifting. Because of the strong ability to absorb solar radiation, dust can heat up the air in the mid-to-upper troposphere by an "elevated heat pump" (EHP) effect (Lau et al., 2006), which provides

cyclonic conditions in the lower layer and induces dry convention over the TP (Lau et al., 2006; Ma et al., 2011; Xu et al., 2018). Under these conditions, dust particles can be more easily elevated into upper-troposphere and lower stratosphere, and then continue the long-range transport with a relatively longer lifetime in upper troposphere and lower stratosphere (Huang et al., 2008; Xu et al., 2018). Also, because of the rising of hot air induced by absorbing aerosol (e.g., dust, black carbon) over the TP, more warm

and moist air is draw in Indian sub-continent, leading to an earlier onset of the South Asia summer monsoon and an intensification of the summer rainfall over the Indian subcontinent (Lau et al., 2006).

     Although the long-range transport dust aerosols play an essential role in a rapid climate change over the TP, the characteristics of dust vertical distribution are not well studied, which are crucial to understanding the dust intercontinental transport (Bourgeois et al. 2015, Xu et al., 2018). Using the

Cloud-Aerosol Lidar and Infrared Pathfinder Satellite Observation (CALIPSO) retrievals, Xu et al. (2018) shown that the dust particles were uplifted to the mid-tropospheric over the source region, and then transported eastward. When reached the TP, they were uplifted to the upper-troposphere with dust mass flux of about $10^{10}$ g by dry and deep convections, which were prevailing in spring. Chen et al. (2013) addressed a dust storm occurred in Taklimakan Desert during 26–30 July 2007. They pointed out that a

value of 6.6 Gg/day mass flux was brought into the TP and –5.58, 1.61, and –3.97 $W/m^2$ of dust radiative forcing were produced at the surface, in the atmosphere and at the top of the atmosphere, respectively. However, we are not aware of studies on the dust source contributions at different altitudes over the TP.

     In generally, many studies focused on dust events over the TP (Uno et al., 2004; Huang et al., 2007, 2008; Yumimoto et al., 2009, 2010; Chen et al., 2013). They showed that the dust could made an

important impact on aerosol mass loading through all year with the largest efficiency in spring and summer (Uno et al., 2004; Huang et al., 2007, 2008; Chen et al., 2013). For example, Middleton (1986) showed that the dust coupled with locally generated pollution aerosols from the Arabian Peninsula and Indian sub-continent was transported and loaded into the TP over the southern slop. The strength of the North African and Arabian Peninsula dust had significantly seasonal variability and it was observed to

peak in summer (Liu et al., 2008; Alizadeh-Choobari et al. 2014; Jin et al., 2016). Huang et al. (2007) indicated that dust plumes originated from the nearby Taklimakan desert were lifted to a very high altitude (7–10 km) over the northern slop. Chen et al. (2017) pointed that a dust storm occurred in 18–23



March 2010 brought dust aerosols from Taklimakan Desert dust (up to 0.8 ton/day) and Gobi Desert (up to 3.7 ton/day) into the TP. So far, however, few studies described the dust plumes transported into the

TP on a climatological view. Moreover, although the dust global-scale transport has been observed or simulated, the spatiotemporal characteristics of dust particle size and number are poorly understood, which could affect the precipitation through modifying the cloud formation and distribution over the TP (Huang et al., 2014; Li et al., 2016), further changing the ice clouds and hydrological cycle to influence the lives of people in South and East Asia (Xu et al., 2009; Yao et al., 2012).

In this study, we implement a newly developed tracer-tagging method in Weather Research and Forecasting model with chemistry (WRF-Chem) to characterize the intercontinental transports of dust emitted from different major desert sources and quantify their contributions to the total dust mass loading over the TP (Wang et al., 2014; Hu et al., 2016, 2019). A 6-year (2010-2015) quasi-global simulation is conducted in the updated version of WRF-Chem. Hu et al. (2016, 2019) has shown that model results

could well represent the dust intercontinental transport over East Asia and the Pacific Ocean compared with multiple observations. Here, our purposes are to (1) explore the transport characteristics of dust particles from different major deserts at different altitudes over the TP; (2) describe the dust particle size distributions at different altitudes over the TP; (3) analysis the dust radiative forcing over the TP; (4) compare dust mass and number concentrations with other aerosols at different altitudes over the TP.

The model setup and satellite retrievals will be described in Section 2 and Section 3, respectively. In Section 4, we will evaluate the WRF-Chem simulated dust aerosol optical depth (AOD) at different altitudes over the TP against satellite observations. In Section 5, the main results will be presented, including the transport characteristics of dust from different source regions, the contribution of different major deserts to the dust mass and number at different altitudes over the TP, the dust mass flux import

into and outflow from the TP and the dust radiative forcing. The conclusion is summarized in Section 6.

## 2 Model

### 2.1 WRF-Chem

In this study, the WRF-Chem model is updated at the University of Science and Technology of China (USTC) based on v3.5.1 (Zhao et al., 2013b, 2014; Hu et al., 2016), particularly, including the

capability of quasi-global simulation and tracer source tagging (Hu et al., 2019). It incorporates Model





for Simulation Aerosol Interactions and Chemistry (MOSAIC) aerosol scheme (Zaveri et al., 2008) and the Carbon Bond Mechanism (CBM-Z) photochemical mechanism (Zaveri and Peters, 1999) for simulating the complex photolysis rates and aerosol radiative properties (Fast et al., 2006). Eight size bins are used in the MOSAIC aerosol scheme to describe the aerosol size distributions by using the bin

approach (Fast et al. 2006), which could better represent the spatial distribution of dust number and dust size loading than 3-mode and 4-bin approaches (Zhao et al., 2013b). Meanwhile, all major aerosol components including mineral dust, sulfate ($SO_4^{-2}$), ammonium ($NH_4^+$), nitrate ($NO_3^-$), organic matter (OM), black carbon (BC) and sea-salt are simulated in the model to represent the sophisticated chemistry during the long-range transport. The aerosol physical and chemical processes in MOSAIC include

coagulation, condensation, nucleation, water uptake, and aqueous phase chemistry. The approach of aerosol dry deposition including gravitational settling and turbulent diffusion follows Binkowski and Shankar (1995). Wet removal of aerosols induced by grid- precipitation and resolved stratiform clouds is simulated by the approach of Easter et al. (2004) and Chapman et al. (2009). The function of wavelength is used for aerosols optical properties of each model grid box. The aerosol longwave (LW) and shortwave

(SW) refractive indices use the Optical Properties of Aerosols and Clouds (OPAC) data set, except the dust SW refractive index, which is defined as 1.53+0.003$i$ following Zhao et al. (2010, 2011). Zhao et al. (2013a) implemented the aerosol radiative feedback for LW and SW radiation into WRF-Chem, which had been coupled with the Rapid Radiative Transfer Model (RRTMG) (Mlawer et al., 1997; Iacono et al., 2000).

**2.2 dust emission**

The dust emissions are simulated by the Goddard Chemistry Aerosol Radiation and Transport (GOCART) model (Ginoux et al., 2001), which has been fully coupled with the MOSAIC in WRF-Chem (Zhao et al., 2010). Zhao et al. (2010) and Hu et al. (2016, 2019) pointed that the model could well represent the dust emission distributions and long-range transport processes over the Northern

Hemisphere. In GOCART, the dust emission flux $G$ is calculated as:

$$G = CSs_p u_{10m}^2 (u_{10m} - u_t),$$

where $C$ (μg s² m⁻⁵) is defined as a dimensionless empirical proportionality constant with the value of 1.0 μg s² m⁻⁵ provided by Ginoux et al. (2001), $S$ is soil erodibility calculated by a potential dust source function, $s_p$ is a fraction of mass of each size bin of dust emissions, $u_{10m}^2$ (m/s) is the horizontal wind



speed at 10 m above the surface, and $u_t$ is the wind speed threshold under which dust emissions do not

occur and is a function of surface moisture, air density, and particle size. To match the particle size

distribution in the aerosol scheme, the dust mass fractions are set to 45%, 26%, 6%, 1.5%, 0.2%, 0.02%,

$10^{-4}$%, and $10^{-6}$%, respectively. The dust emissions within the research area are indicated in Figure 1a.

### 2.3 other emissions

The Hemispheric Transport of Air Pollution version-2 (HTAPv2) is used in the simulations as

anthropogenic emissions with a monthly temporal resolution and 0.1° × 0.1° horizontal resolution

(Janssens-Maenhout et al., 2015). The Fire INventory from NCAR (FINN) supplies biomass burning

emissions with 1 km horizontal resolution and hourly temporal resolution (Wiedinmyer et al. 2011), and

the vertically distribution is followed the injection heights suggested by Dentener et al. (2006). The Sea-

salt emission is obtained from Jaeglé et al. (2011) depended on sea surface temperature and Gong et al.

(2003) based on the correction of particles with radius less than 0.2 $\mu$m.

### 2.4 Numerical experiments

Following Hu et al. (2016), a quasi-global WRF-Chem simulation is performed at 1° horizontal

resolution with 360 × 145 grid cells (180° W–180° E, 67.5° S–77.5° N) and 35 vertical layers up to 50

hPa over the period of 2010–2015 (Figure 1b). The lateral meridional boundary conditions and

meteorological initial are obtained from the National Center for Environmental Prediction final analysis

(NCEP/FNL) data at 6 h temporal and 1° horizontal resolution. In order to better modeling aerosol long-

range transport, the nudged method is used to provide a more realistic atmospheric temperature and wind

components (Stauffer and Seaman, 1990). In addition, a tracer-tagging method is used to tag and

explicitly track dust particles during the intercontinental transport. In the model, four dust source regions

are selected, i.e., East Asia (25° N – 50° N and 75° E – 150° E), North Africa (0° N – 40° N and 20° W

– 35° E), North America (15° N – 50° N and 80° W – 140° W), and the elsewhere in the world (dominated

by the Middle East in the Northern Hemisphere, 0 – 50° N and 35° E – 75° E). The dust particles emitted

from independent regions are tagged and explicitly tracked in the model following Hu et al. (2016, 2019)

and Mao et al. (2019), respectively. Also, the physical and advective tendencies of all tagged dust variable

are treated in the same way. Zhao et al. (2013b) pointed that the updated model could better simulate the

distribution of aerosol size and number and gave more detailed information about the horizontal and



vertical distribution of aerosols and the aerosol intercontinental transport process. More detailed information on numerical experiment, included physical and chemical schemes, can be found in Hu et al.

195     (2016).

### 3 Satellite retrievals

#### 3.1 MODIS

The AOD datasets obtained from Moderate Resolution Imaging Spectroradiometer (MODIS) instrument onboard the NASA EOS Terra satellite is used to evaluate model simulations. It provides

datasets with daily and $1^o \times 1^o$ resolution for a period of 2010–2015. In order to qualify better presumably AOD, the "dark target" algorithm is used to retrieve aerosol size parameters and AOD over vegetated and waters lands (Kaufman et al., 1997; Remer et al., 2005), and the "deep blue" algorithm is used to retrieve the AOD over vegetated land and bright targets (Hsu et al., 2006, 2013). The uncertainties in the merged AOD (at 550 nm) from MODIS collection 6 are about 0.11 over global and 0.069 over the North-

East Asia (Sayer et al., 2014) when compared with AErosol RObotic NETwork (AERONET) observations. In this study, the "dark target" AOD over ocean and "deep blue" AOD over land are merged in MODIS collection 6.

#### 3.2 MISR

The AOD from Multi-angle Imaging SpectroRadiometer (MISR) aerosol Level 2 version 23 is

also used. MISR instrument onboard the Terra spacecraft is designed to view the Earth using nine separate cameras. Four spectral bands (centered at 446, 558, 672 and 866 nm) are measured along the orbital track angles with the ranging from $\pm 70.5°$ (Diner et al., 1998). They can provide well retrieval AOD daily products (Level 2) with a spatial resolution of $0.5^o \times 0.5^o$, especially over highly reflective surfaces such as deserts (Martonchik et al., 2002). The uncertainties in MISR AOD (version 23) is about 0.049, which

is more realistic than 0.003 uncertainty in version 22 products (Witek et al., 2017). Here, we compare the MISR AOD at 550 nm from Level 2 with the model results. It is noticed that the model results are sampled in the same time of MISR overpass.

#### 3.3 CALIPSO



The aerosol extinction profiles retrieved by the Cloud-Aerosol Lidar with Orthogonal Polarization (CALIOP) are used in this study. It was launched into a Sun-synchronous orbit onboard CALIPSO satellite on 28 April 2006. The vertical profiles of attenuated backscatter can be acquired during both night and day phase (Winker et al., 2007). The CALIPSO Level-3 profile products, which are retrieved from Level-2 aerosol profile datasets through a quality-screened, are used to evaluate the model. For the CALIPSO aerosol extinction retrievals, the uncertainties are mainly determined by the

lidar ratio and the misclassification of aerosol type (Winker et al., 2009). Yu et al. (2010) reported that the fraction uncertainty of the lidar ratio is about 30%, which could result in an AOD fraction uncertainty of about 50% (for the AOD is about 0.5). Moreover, the dust extinction at 532 nm with the vertical resolution of 60 m (from –0.5 to 12 km) and the spatial resolution of $5^{\circ} \times 2^{\circ}$ (longitude × latitude) is used. In this study, the CALIOP nighttime observations are used to compared with the dust extinction profiles

of model in cloud-free condition, because the CALIOP are influenced by less noises during nighttime than daytime (Winker et al., 2010).

## 4 Evaluation of AOD and dust AOD simulated by WRF-Chem

Figure 2 illustrates the spatial distributions of the observed (MODIS and MISR onboard Terra) and modeled (WRF-Chem simulation) seasonal AOD at 550 nm for the period of 2010–2015. The simulated

AOD is interpolated at 550 nm by using the Ångström exponent from 400 nm and 600 nm results due to the wavelength discrepancy between simulations and observation. Also, we extract and average the simulated AOD at 10:30 LT (local time) to match the MISR and MODIS retrievals. Because of the missing values of satellite retrievals, there are blank areas over the ocean, which are relatively larger in JJA (Hu et al., 2016). The MODIS retrievals show a higher AOD over anthropogenic polluted regions

(i.e., Eastern China and Southern India) and the major desert regions (i.e., Sahara, Arabian, Gobi and Taklamakan desert) than the MISR retrievals and model simulations. This is partly because the MODIS retrievals overestimate the AOD magnitude in semi-arid regions due to the large uncertainties associated with the assumed surface reflectance (Remer et al., 2005; Levy et al., 2013). Relatively, the MISR characterizes a better land surface reflectance and retrieves better quality presumably. Moreover, the

simulation overestimated the AOD over the North African deserts and the northern Pacific with the peak value (about 0.16) in spring (MAM) (Fig. S1). Over the TP, the simulation AOD is smaller than that of



MISR and MODIS, and the difference is about -0.08. Overall, the WRF-Chem simulated mean AOD shows consistent spatial pattern when compared with MISR and MODIS retrievals. The spatial correlation coefficients of AOD are about 0.65 and 0.79 for MODIS and MISR, respectively. The

seasonal variabilities of spatial correlation coefficients between the simulation and MODIS (MISR) are 0.55 (0.62), 0.63 (0.66), 0.63 (0.68), and 0.67 (0.75) for winter (DJF), MAM, JJA, and autumn (SON), respectively. Over the dust source regions, the simulated results are closer to the MISR retrievals with the largest AOD (more than 0.6) with the correlation coefficient of about 0.87. The trans-Pacific transport of East Asian aerosols and trans-Atlantic transport of Saharan dust are also well simulated, which is

consisted with the studies of Hu et al. (2016) and Yu et al. (2010, 2012). The model also captures the seasonal variabilities of AOD over the major desert regions shown in the satellite retrievals. The maximum AOD over North Africa and Middle East appears in JJA, while that over East Asia is in MAM .

Globally, the dust emissions often occur over source regions with the terrain elevation generally below 3 km and then they can be lifted into upper-troposphere (Yu et al., 2008). By using the CALIPSO

retrievals, Xu et al. (2018) studied the dust transport at different attitude and found that the TP had an important impact on global dust long-range transport on the upper-troposphere. In order to evaluate the model results, we compare the spatial distribution of annual dust AOD from WRF-Chem between 0–3 km, 3–6 km, 6–9 km, and 9–12 km with the CALIPSO results (Figure 3). It shows that the model simulations reasonably reproduce the spatial distributions of dust AOD at different altitudes with

correlation coefficients of 0.71 (0-3 km), 0.86 (3-6 km), 0.49 (6-9 km), and 0.32 (9-12 km), respectively. The maximum dust AOD appears around the dust sources between 0–3 km with a peak value of 0.6, in which the overestimate of simulation (more than 0.06) is over the North African deserts and the underestimate (less than -0.06) is over Middle East and India (Fig. S2). Between 3-6 km, the difference have similar spatial distributions in 0-3 km. The positive difference is about 0.02 over the North African

deserts, and the negative difference is about -0.02 over southern Arabian Peninsula, Somali Peninsula and the northern slop of the TP (Fig. S2). Between 6–9 km and 9–12 km, the modeled dust AOD are higher than CALIPSO with the positive difference of 0.001~0.01 (Fig. S2). The reason would be that the CALIPSO nighttime retrievals in cloud-free condition are used, which have higher accuracy than daytime observations (Winker et al., 2009). Besides, the model results are averaged in all day.



## 5 Result

### 5.1 Characteristics of dust transport from different source regions

The TP is surrounded by several large deserts in the world. On the northern side, the major deserts of Taklimakan and Gobi emit mineral dust particles that can be transported into the TP and elevated to a very high altitude, i.e., approximately 4–7 km (Huang et al., 2007) in most years and up to 12 km during some extreme conditions (Liu et al., 2008). Also, the dust particles from deserts on the western side (i.e., Thar desert, Middle East desert) could be brought into the TP by the westerlies (Jin et al., 2016; Lau et al., 2016). Hu et al. (2019) has pointed out that the North African dust mixes with the Middle East dust before importing into East Asia. In order to better describe the impact of dust from different sources over the TP, the horizontal spatial distributions of annual mean dust mass loading emitted from East Asia (Figure 4a), North Africa (Figure 4b) and Middle East (Figure 4c) are illustrated. Clearly, The East Asian dust dominates the northern slope of the TP and even extends to the southern TP. Meanwhile, the dust mass loading decreases with altitude. In 0–3 km, the maximum dust mass loading occurs over the source regions with the value more than 200 mg/m$^2$. In 3–6 km, dust can climb up to the TP and even reach the southern edge of the TP. Over the northern slop, the dust mass loading is about 50 and 10 mg/m$^2$ in 3–6 km and 6–9 km, respectively. In 9–12 km, the dust (5 mg/m$^2$) appears over the TP due to the effect of the TP orography. Compared with other sources, the East Asian desert contributes more than 90% in 0–3 km over the northern slope of the TP. In 3–6 km, the contribution from East Asian dust is dominated the north of 35º N with the contribution of about 80%. However, the contribution decreases to 40% above 6 km because of the mixing of dust particles from North Africa and Middle East (Figure 4b and 4c). It worth noting that there are significant dust plumes over the TP (Figure 3), especially the northeast region. This dust transport can be attribute to the effect of the TP, which could provide more favorably condition (i.e., cyclonic and dry convection) to make dust uplift into upper-troposphere (Xu et al., 2018).

The North African dust can be transport westward across Atlantic Ocean by African Easterly Waves, which is the major global dust long-range transport, and has been addressed by many studied (e.g., Liu et al., 2008; Su and Toon, 2011; Nowottnick et al. 2011; Ben-Ami et al., 2012; Yu et al., 2013). Beyond that, the North African dust could transport into Europe (Park et al., 2005; Lee et al., 2010) and Middle East (Hu et al., 2019). Hu et al. (2019) pointed that the North African dust mixed with the Middle East dust could be transported eastward and imported into East Asia. During the eastward transport, the TP





stands on the way and plays an important role to affect the dust transport and vertical distribution over

305 the TP. Clearly, the TP can affect the dust intercontinental transport and split it into two branches below

6 km (Figure 4b). When above 6 km, the major transport dust appears over the northern branch and this

can be attributed to the easterlies northward shift (Yu et al., 2008). Meanwhile, more wet deposition

induced by Indian summer monsoon rainfall over the southern branch can reduce the dust transport. The

North African dust brought 10 mg/m$^2$ mass loading over the southern slop below 3 km, but the dust can

reach the northern slop with the value of 10 mg/m$^2$ in 6–9 km. Compared with the East Asia, North Africa

dominates the contribution above 6 km with the value of 40%.

Similar to the North African dust eastward transport, the Middle East dust can also be transported

into East Asia (Hu et al., 2019). Over the TP, the dust mass loading is about 50 mg/m$^2$ in 3–6 km, which

are concentrated in both southern and northern slops (Figure 4c). In the higher altitude, the dust can

overpass the TP with 10 mg/m$^2$ in 6–9 km and 5 mg/m$^2$ in 9–12 km. Also, the Middle East dust is

transported over the 30º N to north and this is similar with the North African dust above the 6 km.

Different with North Africa and East Asia, the contribution from Middle East dust is concentered over

the southern slop of the TP, and the value decreases with the altitude. Below 3 km, the maximum

contribution value occurs over the source region (more than 90%). In 3–6 km, the contribution is about

80% over the southern slop, and the value reduces to 60% in 6–12 km.

Figure 5 illustrates the annual mean wind field in the horizontal (denoted by arrows) and vertical

directions (indicated by the colors) at various altitudes from WRF-Chem simulations for the period of

2010–2015. The positive (negative) value of vertical wind velocity is shown an updraft (downdraft). In

0–3 km, the updrafts appear over major dust source regions, i.e., Sahara desert, Arabian desert, Thar

desert, and Gobi and Taklamakan desert. Because of the updrafts, dust particles emit into the atmosphere

over the sources and then transport to downwind regions (Figure 4a, 4b, and 4c). In 3–6 km, a significant

updraft occurs around the TP, especially over the southern slop. Therefore, the dust from Middle East

and North Africa accounts more fraction contribution than the northern slop. In 6–9 km and 9–12 km,

the updraft wind over the TP reach 0.012 m/s, which is greater than the vertical wind velocity over the

surrounding areas. The reason could be attributed to the dynamical (Huang et al., 2008, Liu et al., 2008)

and thermal forcing due to the TP (Lau et al., 2006) on the atmospheric flows. Under these updraft

conditions, the dust particles can be brought into upper-troposphere (~12 km), which is consistent with

the results from CALISPO observation (Liu et al., 2008).





To describe the transport characteristics of dust from various deserts, we analyze the vertical cross-

sections of meridional circulation and zonal dust mass concentration at 75º E, 90º E and 102º E,

respectively (Figure 6). The three longitudes crossed the TP are donated by purple solid lines in figure

4a. Clearly, strong dust mass from East Asia and Middle East is accumulated over the northern and

southern slop of the TP, then lifted up to the TP due to the orographic lifting. The East Asian dust lifts up

to the TP from its northern slop, and the Middle East dust is major from the southern slop. Also, the

strong northwesterly can transport the East Asian dust continuously eastward (Liu et al., 2015) and the

dust mass concentration increases significantly from 75º E to 102º E. The Middle East dust imports into

the TP along with the westerly wind, but the dust mass concentration decrease from 75º E to 102º E. The

vertical-latitude cross-section of the modeled dust mass concentration centered at 90º E suggests that the

dust particles can lift up to 9 km with the concentration value of 2 $\mu g/m^3$, especially for the Middle East

dust (higher than 10 km). Different with the East Asian and Middle East dust, the North African dust has

a peak mass concentration (~4 $\mu g/m^3$) in the 3–8 km (at 90º E) after imported into the TP.

Figure 7 shows the vertical-longitude cross-section at 24º N, 36º N and 38º N of zonal circulation

and meridional dust mass concentration from the WRF-Chem simulation for the period of 2010–2015.

These latitudes are same with Xu et al. (2018) (Figure 4b), which can well represent the dust zonal

transport over the TP. The maximum East Asian dust mass concentration is located in the 80º E – 120º E,

and the North African and Middle East dust are in 20º W – 40º E and 30º E – 80º E, respectively. These

distributions are consistent with their source regions. The dust emitted from East Asia lifts up to the TP

at 38º N with the peak mass concentration of 40 $\mu g/m^3$, then arrives over the TP (at 36º N) with a value

of 2–10 $\mu g/m^3$. After pass through the TP, there are very few dust particles over the southern slop of the

TP. Because the dust from North Africa and Middle East is mainly concentrated over 30º N to north, the

eastward transport of dust particles is not significantly. At the 36º N and 38º N, the dust eastward transport

is larger with two peaks on both sides of 100º E. These peak values are attributed to the updraft wind on

the western side and downdraft on the eastern side in the westerly circulation.

Overall, the East Asian dust is lifted up to the TP over the northern slop and the Middle East dust is

over the southern slop (Figure 4a and 4b). However, the North African dust important into the TP in a

higher altitude (Figure 5). Above 3 km, the westerly wind is effected by the TP, which results in that the

major transport pathway of dust is over the north of 30º N (Figure 4a, 4b and 4c). In the higher altitude

(above 6 km), the major contributor of dust over the TP is the Middle East dust with a value 60%.



Obviously, the dust particles can be brought to 9 km under the TP orographic effect (Figure 5).

**5.2 Dust mass flux inflowed into and outflowed from the TP**

The annual dust mass flux imported into or outflowed from the boundaries of the TP at various altitudes is shown in Figure 8. In order to better describe the dust mass flux imported into or outflow from the TP, we use meridional wind to calculate the dust mass flux at West (75° E) and East (104° E) side with the latitude range of 27° N - 38° N, and use zonal wind to calculate the dust mass flux at North

(38° N) and South side (27° N) with the longitude range of 75° E - 104° E, respectively. The annual dust mass imported into the TP from the West side is dominated by the Middle East dust (26.6 Tg/year), followed by the North African dust (7.8 Tg/year). The Middle East dust mass flux decreases from 12.3 Tg/year (0–3 km) to 1.2 Tg/year (9–12 km) with the altitude. The North African dust mass flux increases from the surface to 9 km with a peak value of 3.3 Tg/year. In 9–12 km, the North African dust mass flux

(1.4 Tg/year) is greater than Middle East (1.2 Tg/year). At East side, the dust is mainly outflowed from the TP with a peak value (8.7 Tg/year) in 6–9 km. Impacted by East Asian dust, the dust mass flux at North side is dominated by the East Asian dust with a peak value (6.2 Tg/year) in 3–6 km, followed by 0–3 km (3.1 Tg/year). In 6–9 km, the North Africa contributes about 1.1 Tg/year, while the East Asia is −0.9 Tg/year (outflow) due to the westerly wind shift to north (Figure 5). In 9–12 km, the East Asian dust

mass flux is −0.5 Tg/year. At South side, the dust is mainly outflow from the TP with the maximum value from Middle East (−10.4 Tg/year) below 6 km. The Middle East (North Africa) mass flux is about −5.0 (−0.5) Tg/year in 0–3 km and −4.2 (−0.6) Tg/year in 3–6 km.

Although most case studies of dust aerosols have been focused on the late spring and early summer over the TP (Huang et al., 2008; Chen et al., 2013; Liu et al., 2015), seasonal variabilities of long-range

transport dust at different altitude are not well understood. Figure 9 shows the model-based estimation of dust annual mass flux between 0–3 km, 3–6 km, 6–9 km and 9–12 km over the TP for the period of 2010–2015. Clearly, the dust mainly imports into the TP below 6 km. In 0–3 km, the peak inflow and outflow are in JJA and MAM, with the flux of 7.2 Tg and 2.1 Tg, respectively. The weakest inflow (outflow) is in DJF, with a magnitude of 0.82 Tg (0.6 Tg) of the mass flux. In 3–6 km, the inflow and

outflow is the greatest in MAM. The maximum value is 8.9 Tg and 4.4 Tg, respectively. Differently, the second greatest inflow is in JJA (5.6 Tg), but the outflow is in DJF (2.7 Tg). The minimum season is SON, with the mass flux of 2.0 Tg (inflow) and 1.5 Tg (outflow). In 6–9 km, the peak value is in MAM





with a mass flux of 4.6 Tg (inflow) and 5.3 Tg (outflow), followed by DJF (2.2 Tg for inflow and 2.4 Tg for outflow). In 9–12 km, the peak value occurs in MAM with the inflow mass flux of 1.8 Tg and the

outflow mass flux of 2.7 Tg, which accounts about 50% and 54% of the annual flux, respectively. Moreover, we can see that the outflow is greater than the inflow above 6 km while the inflow is greater than the outflow below 6 km especially in 9–12 km due to the effect of the TP topographic.

### 5.3 Characteristics of size distribution of dust over the TP

Dust size representation can significantly influence dust mass balance in the atmosphere and dust

dry/wet deposition (Zhao et al., 2013b). In order to better understanding the dust particle characteristics over the TP, the vertical distribution of dust mass *vs.* dust particle size from 0.039 to 10.0 $\mu$m over the TP is analyzed in Figure 10. Obviously, dust mass vertical distribution *vs.* dust size has significant seasonal and altitudinal variabilities. The East Asian dust is dominated in the size range of 1.25~10.0 $\mu$m above 1 km, in which the maximum dust mass concentration (up to 4 $\mu$g/m$^3$) ranges in the size of 2.5~5.0

$\mu$m with a peak in MAM, followed by the JJA. This is consistent with previous studies that Taklimakan Desert has a peak emission rate of dust in MAM (i.e., Huang et al., 2008, 2013; Chen et al., 2013; Liu et al., 2015). The North African dust mass is mainly in the range of 1.25~5.0 $\mu$m, which is smaller than that of East Asia and Middle East. This is likely attributed to the larger particles scavenged easier by dry/wet deposition during dust intercontinental transport, and fine particles can be more easily transported and

reside in the atmosphere. Also, the maximum dust mass concentration is centered at 2.5 $\mu$m and these particles can reach about 9 km in the MAM, followed by the DJF. Similar with the East Asian dust, the Middle East dust mass is concentrated in the size range of 1.25~10.0 $\mu$m below 3 km, in which the maximum mass concentration ranges in the size of 1.25~5.0 $\mu$m with a peak (up to 10 $\mu$g/m$^3$) in MAM, followed by the JJA. Above 3 km, the Middle East dust has similar seasonal and vertical distribution in

different particle size to the East Asian dust, both of which can reach 12 km.

Beside dust mass and AOD, dust number is another critical factor to influence the ice nuclei concentration number (e.g., Phillips et al., 2008; DeMott et al., 2010), and determines the distribution of cloud, radiation, and precipitation (Zhao et al., 2013a). By comparing dust number loading from three different approaches of representing dust size distribution (8-bin, 4-bin, and 3-mode), Zhao et al. (2013b)

pointed that 8-bin approach could better represent the dust number distribution in different size. In this study, we use the MOSAIC aerosol scheme with 8-bin to simulate long-range transport of dust particles.




Figure 11 shows the spatial distribution of dust number loading between 0–3 km, 3–6 km, 6–9 km and 9–12 km. Clearly, the dust number concentration reduces from west to east, but the gradient is smaller than that of dust mass due to a faster removal rate of mass in general (Zhao et al., 2013b). Between 0–3 km, the greatest dust number appears over the source regions (i.e., Sahara desert, Arabian desert, Thar desert, Gobi and Taklamakan desert) with a value of 0.08 $\mu m^{-2}$. Between 3–6 km, dust number loading reduces and a value of ~0.008 $\mu m^{-2}$ appears over the TP. The dust number loading increases (~0.02 $\mu m^{-2}$) due to the orographic effect, which is consistent with the dust AOD distribution in 6–9 km (Figure 3). In 9–12 km, there is a peak in dust number over the north of the TP with a value of ~0.008 $\mu m^{-2}$.

Figure 12 shows the vertical distribution of dust number *vs.* particle size from 0.039 to 10.0 $\mu m$ over the TP. It suggests that dust number has similar seasonal variability with the dust mass *vs.* dust particle size. However, the size range is different for the dust from various sources over the TP. The East Asian and North African dust number are dominated in the range of 0.156~1.25 $\mu m$, but the Middle East dust number is fell in a broader range of 0.078~2.5 $\mu m$. Compared with the dust mass, dust number loading over the TP is mainly in smaller particle size. This is likely due to easier suspension and long-range transport of smaller dust particles (Chin et al., 2007). The East Asian dust number concentration over the TP is mainly in 2~8 km with a peak (~0.4 $mm^{-3}$) in MAM, and has the smaller dust number concentration compared with other sources. While the North African and Middle East dust number concentration have a broader vertical distribution, even reach to 12 km. For the Middle East dust number, there is a significant peak value occurred (more than 1 $mm^{-3}$) below 4 km, which is larger than other sources. This result implies that the dust from Middle East could contribute more particles in the dust number, which could influence the regional cloud and precipitation.

**5.4 Dust radiative forcing over the TP**

Figure 13 shows the spatial distribution of annual mean SW, LW, and net (SW + LW) direct radiative forcing of dust at the TOA, in the atmosphere (ATM) and at the surface (SFC) from the WRF-Chem simulations averaged for 2010–2015. Over the TP, the dust SW radiative forcing is negative at TOA with a value around –0.5 W/m², which is far less than the value over the desert regions due to the high dust albedo and absorbing capability (Zhao et al., 2011). Because of dust LW absorption, the dust LW radiative forcing at TOA is warming with a positive value of 0.5 W/m². Overall, the dust net radiative forcing at TOA have a cooling effect. In the atmosphere, the dust warms the atmosphere through absorbing SW and





cools the atmosphere by increasing atmospheric outgoing LW radiation. Dust produces a net warming effect with a value of 0.5 W/m² over the TP. At the surface (SFC), the dust induces a cooling effect by reducing downward SW radiation and a warming effect by trapping the LW radiation emitted from surface. Moreover, the dust net radiative forcing is negative and cooling the surface. The seasonal and

annual mean SW, LW, and net radiative forcing of dust simulated by WRF-Chem over the TP are summarized in Figure 14. The dust significantly cools the TOA and SFC, and warms the atmosphere with a peak effect in MAM, because of larger dust mass concentration in this season over the TP (Hu et al., 2019). Also, the annual dust SW radiation forcing is –1.40 W/m² (TOA), 0.67 W/m² (ATM) and –2.08 W/m² (SFC), which is far larger than dust LW radiation forcing (0.13 W/m² (TOA), –0.26 W/m² (ATM)

and 0.39 W/m² (SFC)). Overall, the dust-induced change of net radiation forcing is –1.28 W/m² at the TOA, 0.41 W/m² in the ATM and –1.68 W/m² at the SFC over the TP. The annual mean radiative effect of dust is less than the results from Chen et al. (2013), which investigated the dust radiative forcing under an intense dust storm event.

### 5.5 Comparison between dust and other aerosols at different altitudes over the TP

Figure 15 shows the vertical distribution of seasonal AOD, aerosol mass and aerosol number over the TP. Significantly, AOD, aerosol mass and aerosol number decrease with the attitude. In 0–3 km, the peak AOD (0.24) is in JJA, but the peak aerosol mass (88 $\mu$g/m²) and number (7.2 $\mu$m$^{-2}$) are in MAM. While, above 6 km, the effect of aerosols has a peak in MAM. Also, the aerosol mass and number have a larger decrease from surface to 6 km. In the 6–12 km, the weaker decreasing indicates that dust and

sulfate particles have less decrease, which account for more than 50% in total aerosols (Figure 16). In the higher altitude (9–12 km), there is no obvious seasonal changes for the aerosols.

In order to explore the relative contribution of dust, the fraction from dust and other components is shown in Figure 16. Clearly, the fraction of dust AOD shows an increasing trend with altitude. The dust AOD fraction has significant seasonal variabilities in different altitude with a peak in MAM. For aerosol

mass, the dust fraction is more than 50% in MAM (57%) and JJA (65%) due to the maximum dust intercontinental into the TP in 0–3 km. Above 3 km, the dust mass contribution fraction is more than 55%. The maximum contribution value occurs in 6–9 km in all seasons, expect JJA (occurred in 3–6 km). However, the dust number contribution can be ignored. The major components for aerosol number are sulfate, OM and other aerosols. The sulfate number contribution increases with the altitude, but the OM



decreases. In 0–3 km, the OM contributes the greatest number with the value more than 40%. In 3–9 km, other aerosols contribute about 40% number. In 9–12 km, the sulfate contributes more than 45% number.

**6 Conclusions and Discussion**

In this study, we conduct a quasi-global WRF-Chem simulation and implement a newly developed tracer-tagging method, which divide the global deserts into four source regions and explicitly on-line

track the emission, transport and deposition of the dust particles. Using this method, we investigate dust source, intercontinental transport and radiative forcing at the different altitudes over the TP for the period of 2010–2015. The main conclusions are summarized below:

1. The spatial and seasonal variability of AOD can be well capture by the model simulation. The observation and simulated results show that the maximum AOD appeared over North Africa, Middle East and Central Asia in JJA, but in MAM over East Asia. Also, the model has a reliable

representation of spatial distribution of dust AOD at different altitudes compared with the CAIPSO retrievals. The maximum dust AOD appears around the dust sources with a peak value of 0.6.

2. Lifted by updrafts, dust particles are emitted into the atmosphere over the major desert regions, and then transport to downwind regions. The East Asian dust transports southward and climbs

up to the TP over the northern slop with the mass loading from 50 (at 3 km) to 5 (at 12 km) mg/m$^2$. The North African dust transports eastward under the westerly wind, mixes with the Middle East dust, and then imports into East Asia. However, the TP stands on the way and splits the dust plumes into two branches. The North African dust imports into the TP over the northern

slop with the mass of 10 mg/m$^2$ below 6 km. While, the Middle East dust concentrates over the southern slop with the dust mass loading of ~50 mg/m$^2$ below 6 km, and overpasses the TP with mass loading of 5–10 mg/m$^2$ above 6 km. Overall, the East Asia contributes more dust mass over the northern slop, and the Middle East contributes more dust mass over the southern slop. In the higher altitude (above 6 km) over the TP, the major contributor is Middle East with a

value of 60%.

3. On an annual basis, the Middle East dust and North African dust mainly imports into the TP at the West side with the mass flux of 26.6 and 7.8 Tg/year, while the East Asian dust imports into



the TP at North side with the mass flux of 8.0 Tg/year. The mass flux of different source dust outflow the TP from the East and South side is –5.4 (East Asia), –6.2 (North Africa), –6.4

(Middle East) Tg/year, and –0.2 (East Asia), –1.3 (North Africa), –9.7 (Middle East) Tg/year, respectively. The greatest dust mass flux imported into and outflowed from the TP is in 3–6 km. Also, the imported and outflowed dust mass flux are mainly occurs in MAM. The peak values are 8.9 Tg (inflow) and 4.4 Tg (outflow) in 3–6 km, and 6.4 Tg (inflow) and 8.0 Tg (outflow) above 6 km.

4.   The dust can be brought into 9 km under the effect of the TP orography. Inside, the East Asian dust mass is dominated in the size range of 1.25~10.0 $\mu$m in spring (MAM), followed by the JJA. While, the North African and Middle East dust mass are mainly in the range of 1.25~5.0 $\mu$m and the dust particles can reach to 12 km, but the Middle East dust take more mass fraction. For the dust number, the East Asian and North African dust are dominated in the range of

0.156~1.25 $\mu$m, but the Middle East dust number is fell in 0.078~2.5 $\mu$m. Relatively, the East Asian dust number over the TP is mainly in 2~8 km, while the North African and Middle East dust number are broader, even reaches to 12 km.

   5.   For the radiative forcing, dust produces the annual mean SW, LW and net (SW + LW) radiation forcing is –1.40, 0.13, and –1.27 W/m$^2$ at the TOA, 0.67, –0.26 and 0.41 W/m$^2$ in the

atmosphere, and –2.08, 0.39 and –1.69 W/m$^2$ at the surface over the TP, respectively. For the aerosol mass, the dust contribution fraction more than 50% in MAM (57%) and JJA (65%) due to the maximum dust intercontinental into the TP in 0–3 km. Above 3 km, the dust mass contribution fraction more than 55% with a peak contribution in 6–9 km. However, there is smaller contribution for total aerosols dust number compared with sulfate, OM and other aerosol

particles in the column. For the aerosol optical characteristics, the fraction of dust AOD has an increasing trend with altitude, while the dust AOD is decreasing.

       The WRF-Chem quasi-global simulation not only captures the spatial and seasonal variabilities of dust long-range transport but also tags the dust particle loading into the TP from different major deserts over North America, East Asia, North Africa and the elsewhere in the world (mainly for Middle East).

Also, this study explores the dust particle size and number distribution over the TP. The dust from Gobi and Taklimakan and desert transport southward and is lifted up to the TP from northern slop. While, dust from North Africa and Middle East transports eastward and then lifts up to the TP from both the northern





and southern slops. Under the effect of high TP topography, the dust particles can reach upper-troposphere (above 8 km) and even stratosphere, which can provide a pathway for dust into the upper-

troposphere (above 9 km). This study provides a climatological view about the intercontinental transport characteristics of dust from different source regions over the TP. Our results are important to understand the source contribution of the dust over TP and the dust belt formation over the Northern Hemisphere. However, the impact of dust aerosols on cloud properties and precipitation are not discussed, which is an important issue over the TP. Dust–cloud–precipitation interactions will be investigated in future.




**Code availability**

The release version 3.5.1 of WRF-Chem can be download from http://www2.mmm.ucar.edu/wrf/users/download/get_source.html. The updated model is available
to contact the first author (huzy@lzu.edu.cn). Also, the code modifications will be incorporated the release version of WRF-Chem.

**Author contributions**

Zhiyuan Hu and Chun Zhao conducted the quasi-global simulations. Zhiyuan Hu performed the analyses, wrote the paper and coordinated the paper. All authors contributed to the final version of the paper.

**Acknowledgements**

This research was supported by the National Natural Science Foundation of China (No. 41805116 and No. 41775146), the Fundamental Research Funds for the Central Universities lzujbky-2019-pd05, lzujbky-2018-49 and lzujbky-2019-kb02, Innovative Research Groups of the National Science Foundation of China (Grant No. 41521004) and Strategic Priority Research
Program of Chinese Academy of Sciences, (Grant No. XDA20060103).



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



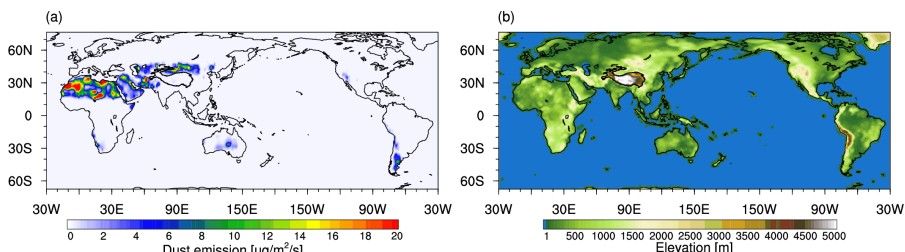

**Figure 1.** Spatial distribution of (a) dust emission (μg/m$^2$/s) and (b) elevation (m) simulated in WRF-Chem for the period of 2010–2015.



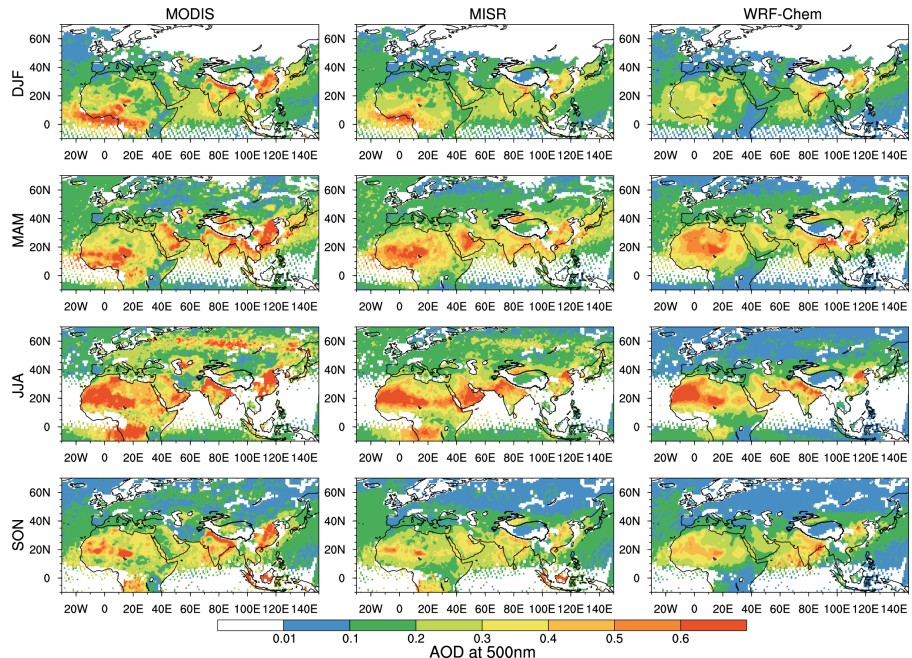


**Figure 2.** Spatial distributions of seasonal mean AOD at 550 nm from the retrievals of MODIS and MISR onboard Terra and the WRF-Chem simulation for the period of 2010–2015. The daily results from MISR, MODIS, and WRF-Chem are only sampled for averaging when all of them have valid values at the same location and time.




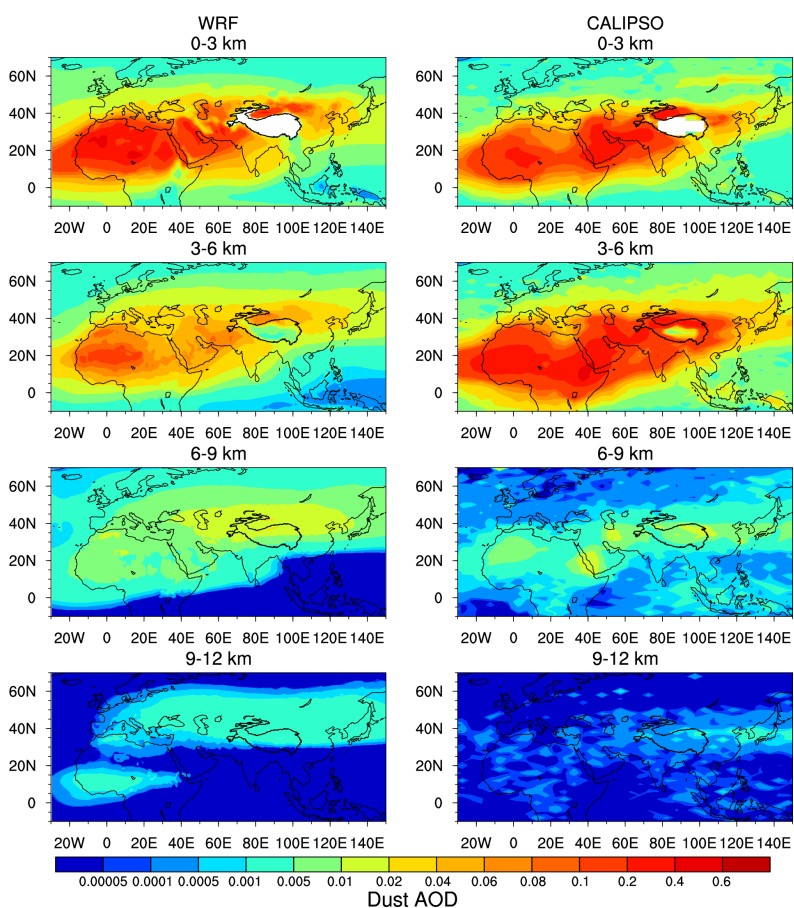

**Figure 3.** Spatial distributions of annual mean dust AOD from CALIPSO and WRF-

Chem between 0–3 km, 3–6 km, 6–9 km, and 9–12 km for the period of 2010–2015.

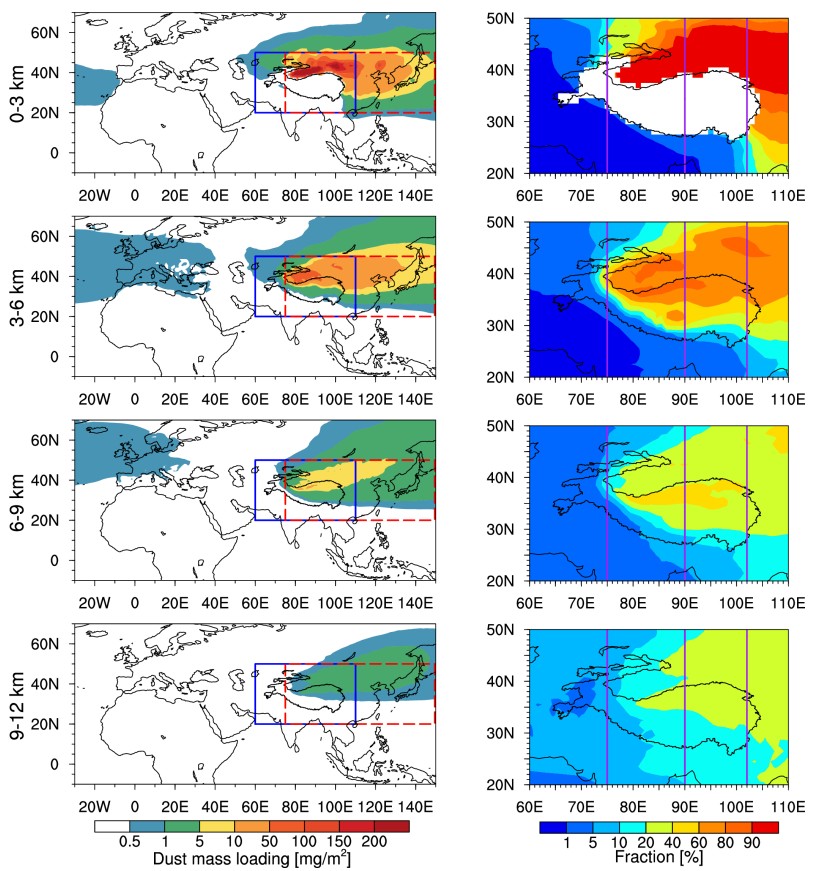

**Figure 4a.** Spatial distributions of (Left) annual mean dust mass loading (mg/m²)

emitted from East Asia between 0–3 km, 3–6 km, 6–9 km, and 9–12 km and (Right)

the corresponding fractions compared with total dust mass loading over TP. The box

in red dotted lines shows the dust source regions, while the box in blue solid lines

shows the TP regions in this study. The purple solid lines are at 75º E, 90º E and 102º

E for the dust height–latitude cross sections.



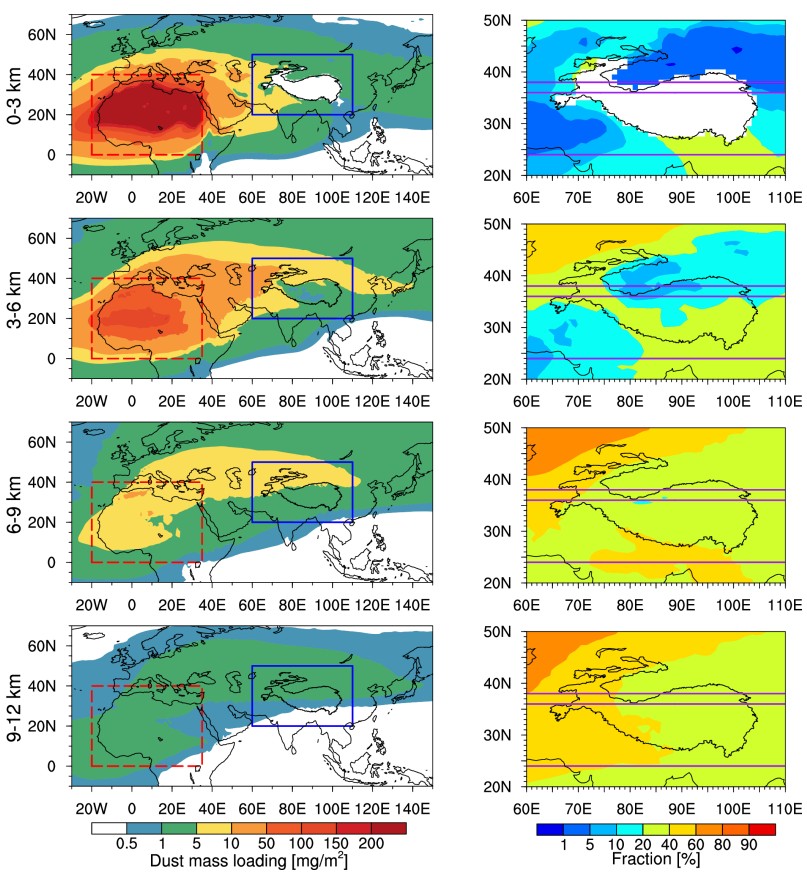

**Figure 4b.** Same as figure 4a, but for dust emitted from North Africa. The purple

solid lines are at 24° N, 36° N and 38° N for the dust height–longitude cross sections.




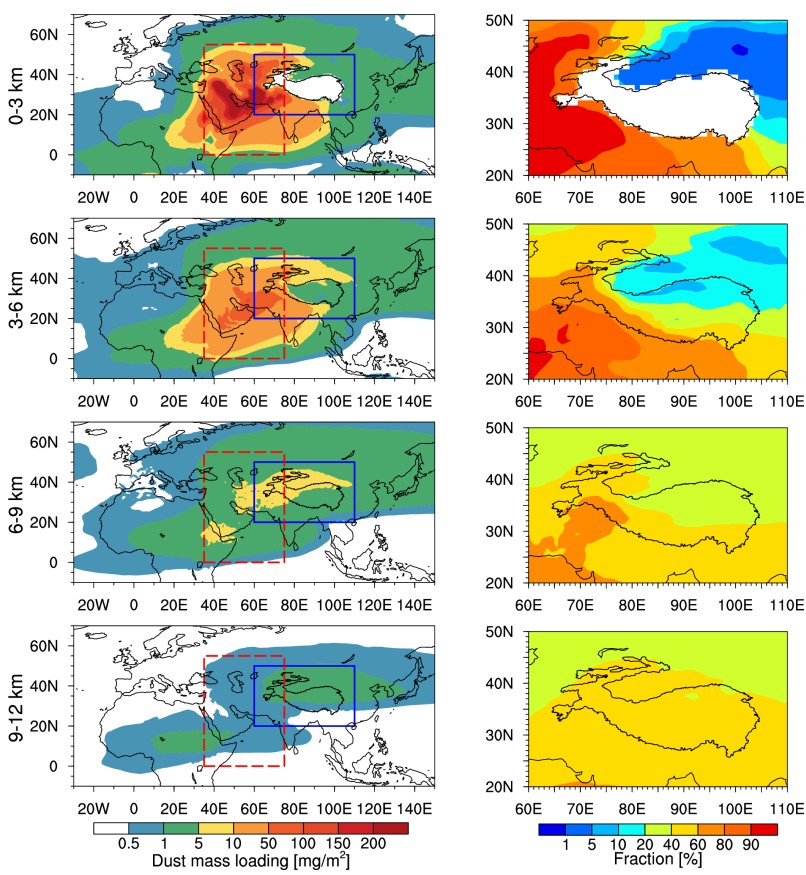

**Figure 4c.** Same as figure 4a, but for dust emitted from the Middle East.

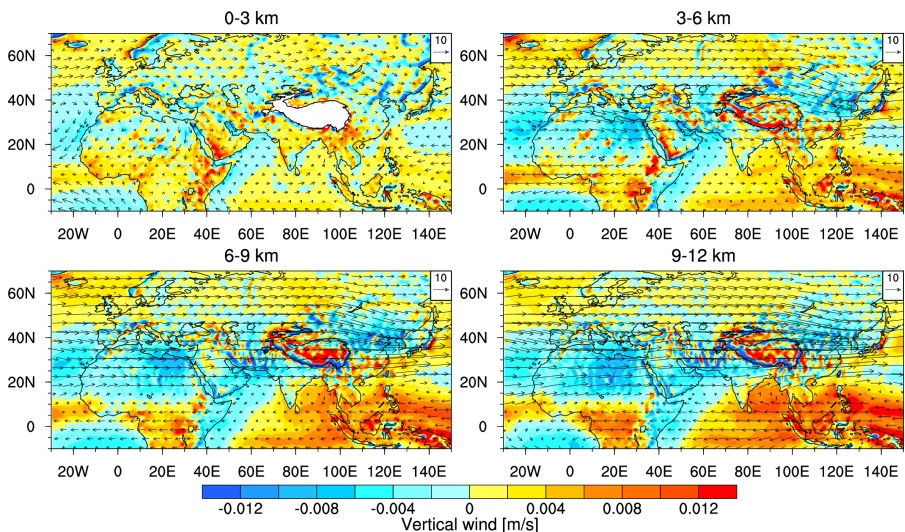


**Figure 5.** Annual mean wind field between 0–3 km, 3–6 km, 6–9 km, and 9–12 km from WRF-Chem simulations for the period of 2010–2015. The arrows indicate the horizontal wind (m/s). The colors indicate the vertical wind velocity (m/s) and positive values are for updrafts and negative values are for downdrafts.


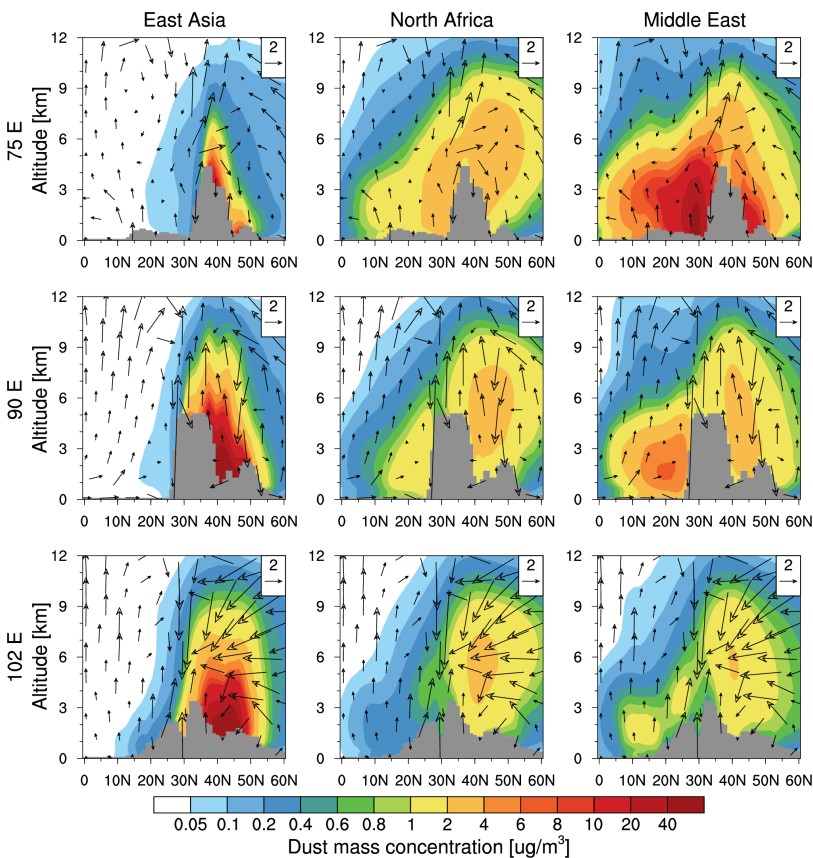

**Figure 6.** Vertical cross-sections of meridional circulation (m/s and scale factor 300 applied for vertical wind) and zonal dust mass concentration (μg/m³) emitted from East Asia, North Africa and Middle East at 75° E, 90° E and 102° E.

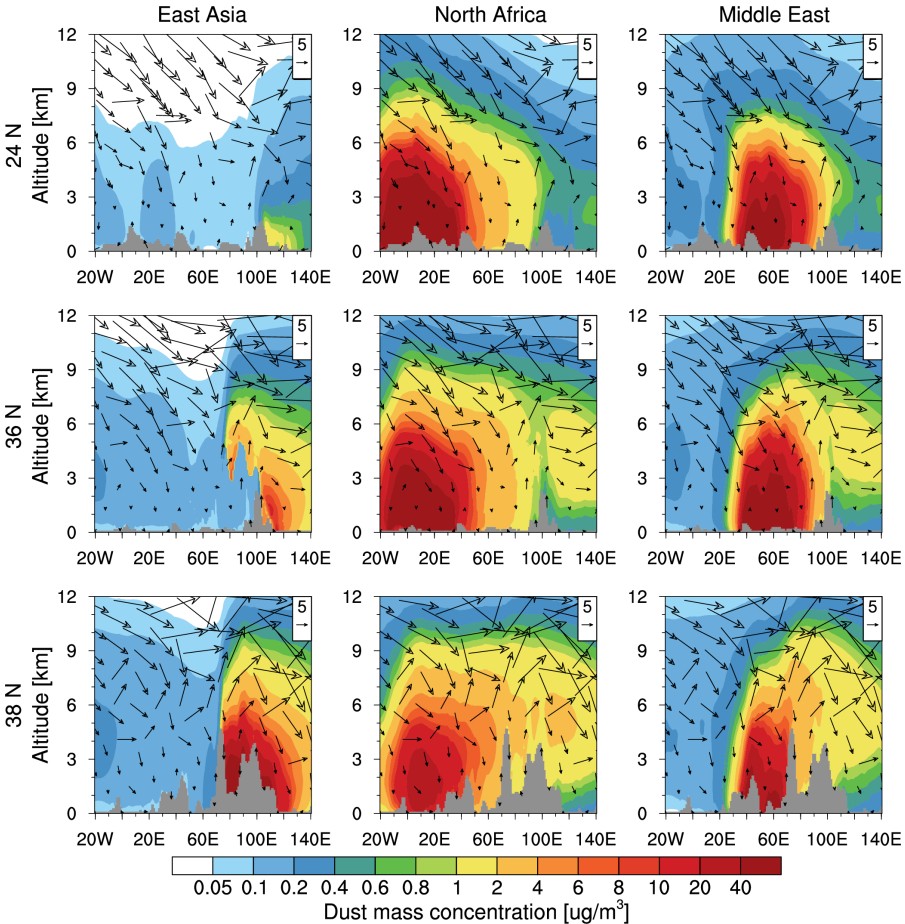

**Figure 7.** Vertical cross-sections of zonal circulation (m/s and scale factor 300 applied

for vertical wind) and meridional dust mass concentration ($\mu g/m^3$) emitted from East

Asia, North Africa and Middle East at 24° N, 36 ° N and 38° N.


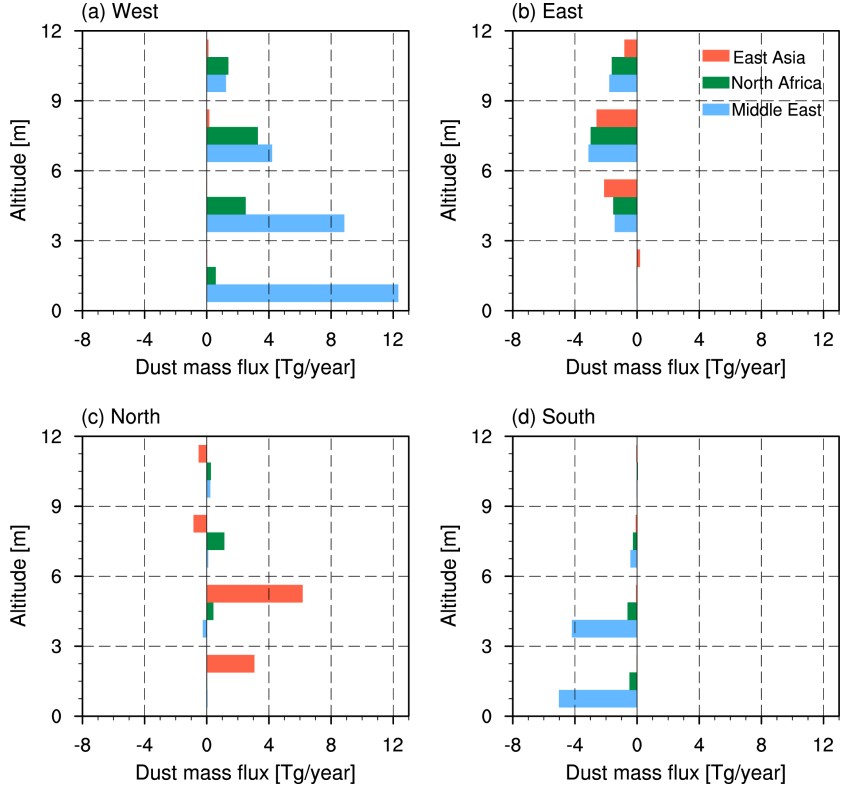

**Figure 8.** Model-based estimation of annual dust mass flux (Tg/year) at the West (a) and East (b), North (c) and South (d) side of the Tibetan Plateau, respectively. The latitude ranges in 27º N – 38º N and the longitude ranges in 75º E – 104º E. The west and east imported mass flux are based on the zonal wind, and South and North are based on the meridional wind. Also, the values are negative for imported and positive for outflowed.



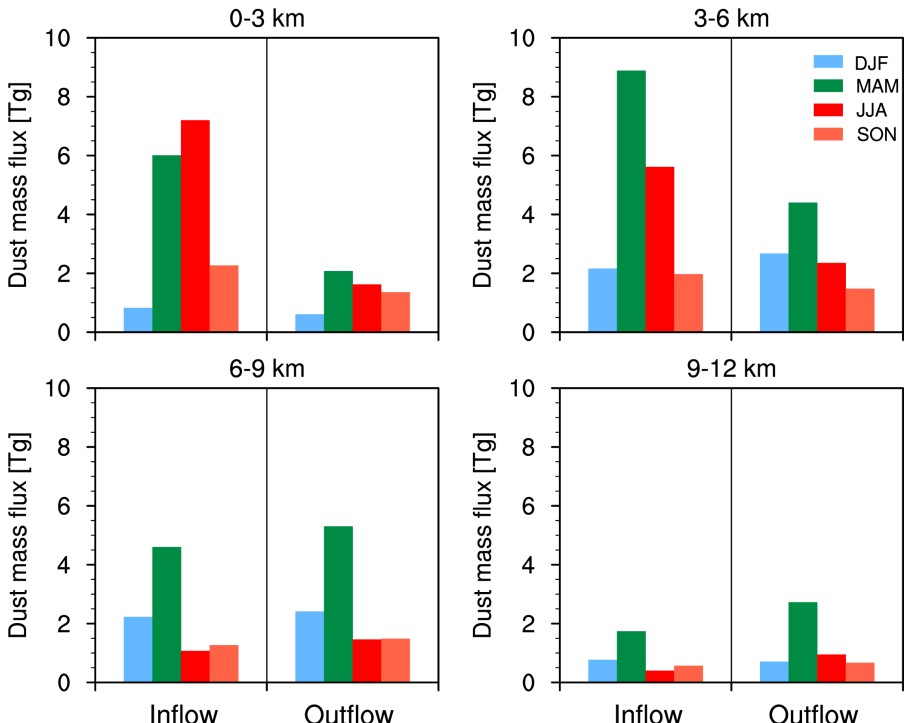

**Figure 9.** Model-based estimation of annual dust mass flux (Tg) inflow and outflow

from the TP between 0–3 km, 3–6 km, 6–9 km and 9–12 km from the WRF-Chem

simulations averaged for 2010–2015.



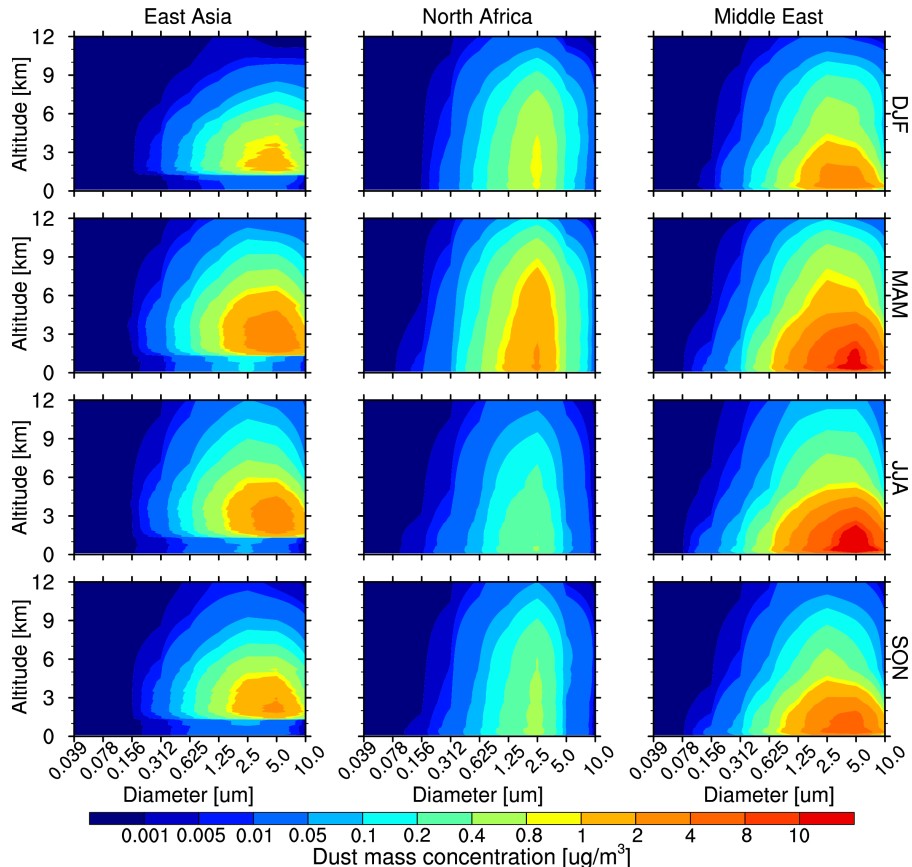

**Figure 10.** Vertical distribution of seasonal dust mass concentration ($\mu$g/m³) over the TP within various particle size sections emitted from East Asia, North Africa and Middle East averaged for 2010–2015.





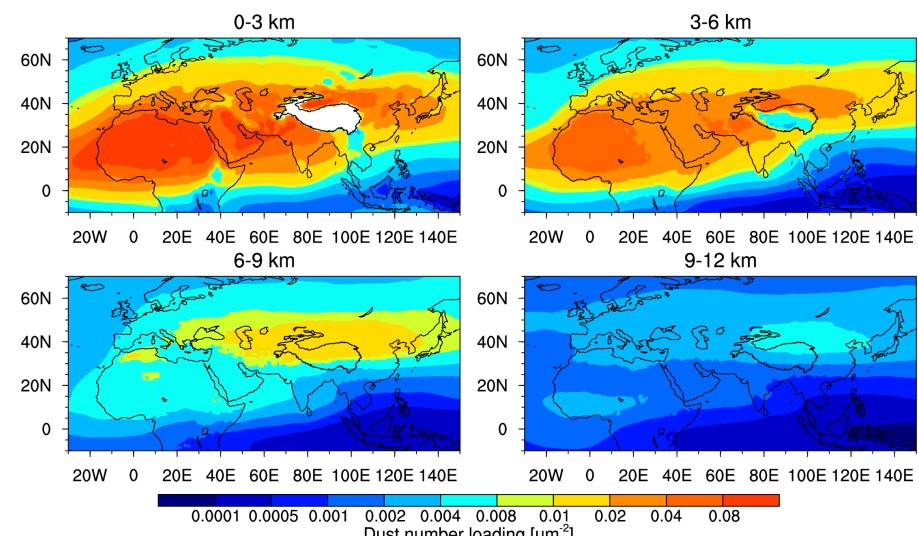


**Figure 11.** Spatial distributions of annual mean dust number loading ($\mu$m$^{-2}$) from WRF-Chem between 0–3 km, 3–6 km, 6–9 km, and 9–12 km for the period of 2010–2015.






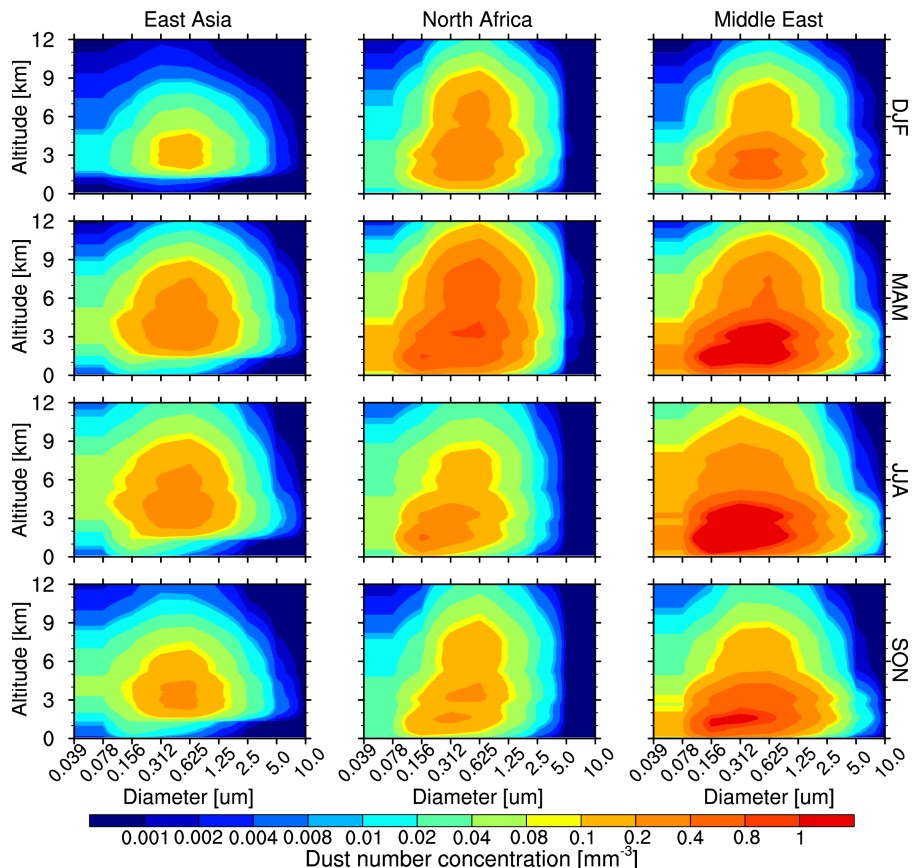

**Figure 12.** Vertical distribution of seasonal dust number concentration (mm$^{-3}$) over the TP in different particle size emitted from East Asia, North Africa and Middle East averaged for the period of 2010–2015.






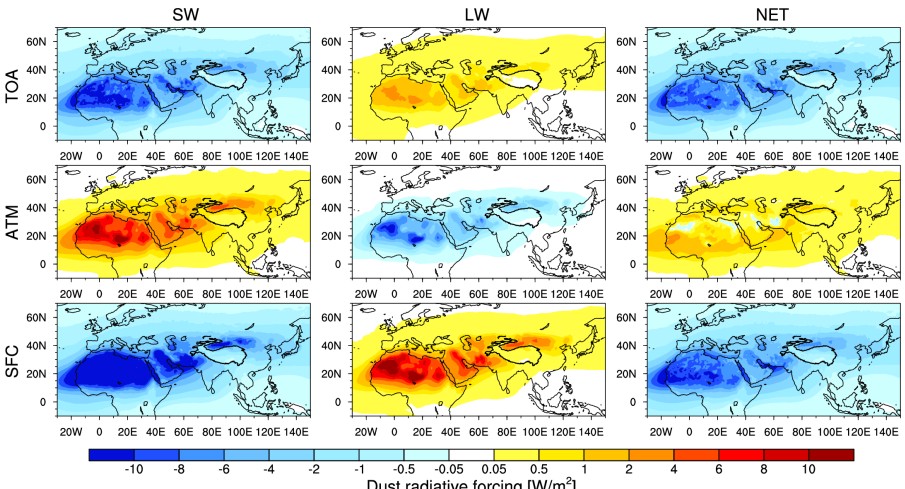

**Figure 13.** Spatial distribution of seasonal mean SW, LW, and net (SW + LW) direct

radiative forcing (W/m$^2$) of dust at the TOA (top panels), in the atmosphere (ATM:

middle panels) and at the surface (SFC: bottom panels) under all-sky conditions from

the WRF-Chem simulation averaged for 2010–2015.




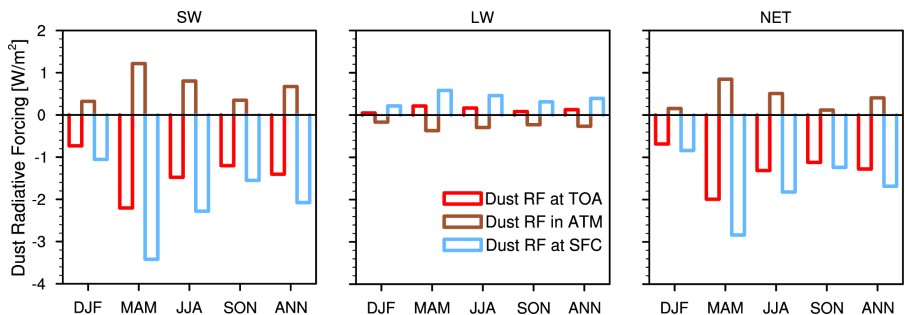

**Figure 14.** Seasonal and annual mean SW, LW, and net (SW + LW) radiative forcing

(W/m$^2$) of dust simulated by WRF-Chem for the period of 2010–2015 over the TP.

Negative values represent downward radiation.



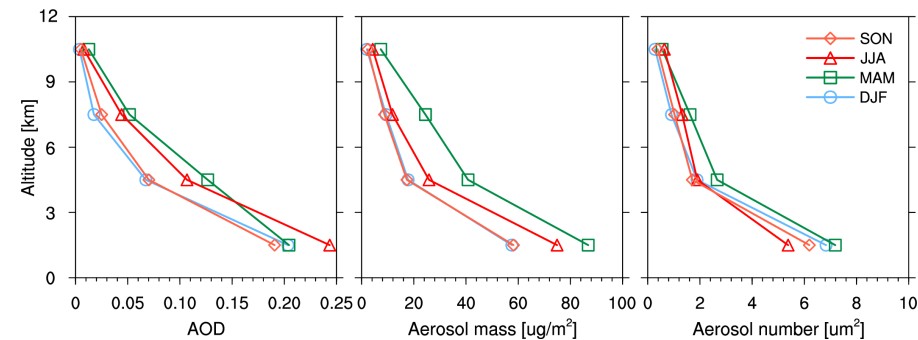


**Figure 15.** Vertical distribution of seasonal AOD, aerosol mass loading ($\mu$g m$^2$), and

aerosol number ($\mu$m$^2$) over the TP from WRF-Chem simulation averaged for the period

of 2010–2015.






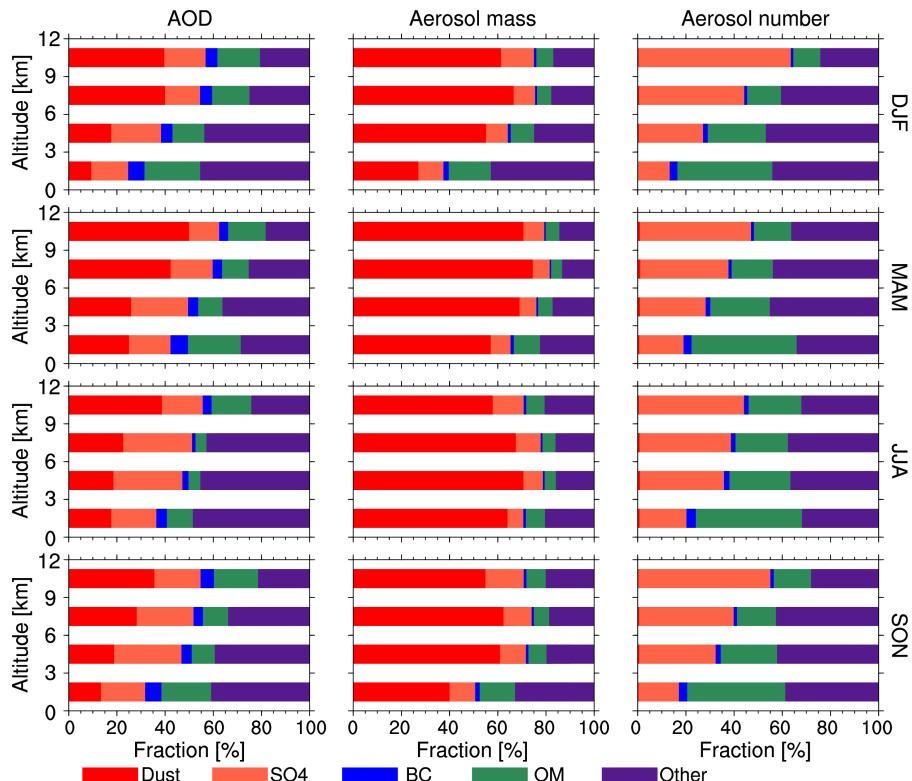

**Figure 16.** Fractions of seasonal aerosol composition in total AOD (left), aerosol mass concentration (middle), and aerosol number (right) in four separate vertical layers: 0–3 km, 3–6 km, 6–9 km and 9–12 km from the WRF-Chem simulation averaged for the

period of 2010–2015 over the TP.