# Peer review of "Modeling dust sources, transport, and radiative effects at different altitudes over the Tibetan Plateau"

_Atmospheric Chemistry and Physics, 2019_

## Referee Comment (RC1) · Anonymous Referee #2 · 29 Aug 2019

General comments: In this work, the authors conduct a quasi-global WRF-Chem simulation and implement a tracer-tagging method to investigate dust source, intercontinental transport and radiative forcing at the different altitudes over the Tibetan Plateau (TP) for the period of 2010–2015. The main conclusions summarized briefly below are reasonable. (1) The model has a reliable representation of spatial distribution of dust AOD (DOD) at different altitudes compared with the CAIPSO retrievals. (2) The East Asia contributes more dust mass over the northern slope, and the Middle East contributes more dust mass over the southern slope. In the higher altitude (above 6 km) over the TP, the major contributor is Middle East with a value of 60%. (3) The East Asia dust number over the TP is mainly in 2∼8 km, while North African and Middle East dust

number is broader, even reaches to 12 km. (4) For the radiative forcing, dust produces the annual mean SW, LW and NET (SW + LW) radiation forcing is −1.40, 0.13, and −1.27 W/m2 at the TOA, 0.67, −0.26 and 0.41 W/m2 in the atmosphere, and −2.08, 0.39 and −1.69 W/m2 at the surface over the TP, respectively. As stated by the authors in Section 6. This study provides a climatological view about the intercontinental transport characteristics of dust from different source regions over the TP, and these results are important to understand the source contribution of the dust over TP and the dust belt formation over the Northern Hemisphere. Overall, the model, method, and data utilized are suitable for this topic, and the context is basically well organized. However, the scientific importance need be further clarified because the dust number accounts for smaller contribution for total aerosols number compared with sulfate, OM and other aerosols in the column over TP. People might be more interested in to what a degree the mineral dust can affect the in situ and surrounding precipitation, diabatic heating, and cloud.

Specific comments: 1. The references in the text should not be limited in the aerosol community, especially for the climate effect of the TP in terms of dynamical and thermal effects. 2. Abstract. "The East Asian dust trasnports southward and is lifted up to the TP". This statement seems to be misleading because East Asia area defined here, i.e., ($25°$ N − $50°$ N and $75°$ E − $150°$ E) covers a large domain with the TP itself included. According the context, it might mainly be Gobi and Taklamakan. A more specific description is needed. 3. Introduction. "As the highest (about 4000 m) and largest plateau in the world, the Tibetan Plateau (TP)...", the largest plateau is not true. 4. Introduction. The conception of "elevated heat pump" can be traced to some earlier literatures. 5. Caption of Fig. 1. It should be climate mean spatial distribution. 6. Section 4. Evaluation of AOD and DOD simulated by WRF-Chem. "Between 6–9 km and 9–12 km, the modeled DOD are higher than CALIPSO. The reason would be that the CALIPSO nighttime retrievals in cloud-free condition are used, which have higher accuracy than daytime observations (Winker et al., 2009). Besides, the model results are averaged in all day." This explanation for model bias seems to be questionable, is

it possible related to circulation condition? How about the nighttime results for model? Also, it's better to put CALIPSO results in the left panels of Fig. 3. 7. The TP domain plotted in Fig. 4 is too large, with many surrounding plains included such as north Indian subcontinent and north Indo-China Peninsula. 8. Fig. 5. The annual mean horizontal and vertical wind field may neglect the remarkable seasonality of TP and adjacent monsoon regions. Particularly for the ascending at 3-6 km but descending above it in the south fringe of the TP. 9. Section 5.4 Dust radiative forcing over the TP. "the dust LW radiative forcing at TOA is waring with a positive value of 0.5 W/m2". Waring is a typo. 10. Fig. 9. The average altitude of TP is 4 Km, how can get the results at 0-3 Km? 11. Fig. 10 and 12 can be merged in one. 12. Section 6 Conclusions and Discussion. "the dust particle can reach upper-troposphere (above 8 km) and even stratosphere, which can provide a pathway for dust into the upper-troposphere (above 9 km)." This expression is strange and hard to follow. 13. Caption of Fig. 13. "Spatial distribution of seasonal mean. . .". For which season the authors mentioned here.

---

## Referee Comment (RC2) · Anonymous Referee #3 · 21 Nov 2019

This study provides long-term quasi-global simulations of dust aerosols and investigates the dust source contribution and transport over the Tibetan Plateau (TP). By a tracer-tagging technique, the dust emitted from different deserts are identified and the contributions of dusts from the global major deserts to the TP are quantitively estimated. Furthermore, the dust radiative effects over the TP are also studied. The manuscript is well organized and written. Therefore, I recommend publishing it once the following comments are addressed.

1. The simulation is conducted with the CBMZ-MOSAIC scheme, please briefly introduce how the anthropogenic gas and aerosol emissions form the HTAPv2 are mapped

to the CBMZ-MOSAIC species. Otherwise, it is difficult for readers to reproduce the experiment. 2. Could you briefly explain the tracer-tagging technique used in the study without reader to refer to other reference. 3. How the dust radiative forcing is calculated in this study? Is it the double call of the radiative module or other method?

---

## Author Response (AR1)

We thank the two anonymous referees for their valuable comments and constructive suggestions on the manuscript. Below, we explain how the comments and suggestions are addressed point-by-point and make note of the revision that we made in the manuscript.

**Anonymous Referee #2**

*General comments:*

- *In this work, the authors conduct a quasi-global WRF-Chem simulation and implement a tracer-tagging method to investigate dust source, intercontinental transport and radiative forcing at the different altitudes over the Tibetan Plateau (TP) for the period of 2010–2015. The main conclusions summarized briefly below are reasonable. (1) The model has a reliable representation of spatial distribution of dust AOD (DOD) at different altitudes compared with the CAIPSO retrievals. (2) The East Asia contributes more dust mass over the northern slope, and the Middle East contributes more dust mass over the southern slope. In the higher altitude (above 6 km) over the TP, the major contributor is Middle East with a value of 60%. (3) The East Asia dust number over the TP is mainly in 2~8 km, while North African and Middle East dust number is broader, even reaches to 12 km. (4) For the radiative forcing, dust produces the annual mean SW, LW and NET (SW + LW) radiation forcing is –1.40, 0.13, and –1.27 W/m2 at the TOA, 0.67, –0.26 and 0.41 W/m2 in the atmosphere, and –2.08, 0.39 and –1.69 W/m2 at the surface over the TP, respectively. As stated by the authors in Section 6. This study provides a climatological view about the intercontinental transport characteristics of dust from different source regions over the TP, and these results are important to understand the source contribution of the dust over TP and the dust belt formation over the Northern Hemisphere. Overall, the model, method, and data utilized are suitable for this topic, and the context is basically well organized. However, the scientific importance need be further clarified because the dust number accounts for smaller contribution for total aerosols number compared with sulfate, OM and other aerosols in the column over TP. People might be more interested in to what a degree the mineral dust can affect the in situ and surrounding precipitation, diabatic heating, and cloud.*

**Response:** We thank the reviewer for a detailed review. We agree that the dust number has smaller contribution to total aerosol number compared with anthropogenic aerosols in the column over the Tibetan Plateau (TP). However, the dust number could not been ignored, because it is an important factor in dust climatic impact over the TP. Zhao et al. (2013) had pointed that dust size distributions and dust number could significantly change dust

radiative forcing. In order to make the scientific importance more clear, we add some discussion on page 5 line 131-138 as: "Moreover, although the dust global-scale transport has been studied using both observations and model simulations, the spatiotemporal characteristics of dust particle size and number and their associated climatic impact are poorly understood (Zhao et al., 2013). The dust number could be used to determine the cloud condensation nuclei and/or ice nuclei (e.g., Liu and Penner, 2005; DeMott et al., 2010) and dust particle deposition (Zhao et al., 2013). Consequently, this could affect the precipitation through modifying the cloud formation and distribution over the TP (Huang et al., 2014; Li et al., 2016), and further change the hydrological cycle and influence the lives of people in South and East Asia (Xu et al., 2009; Yao et al., 2012).". Besides, we also add some discussion on page 22 line 605-607 as: "Also, this study explores the dust particle size and number distribution over the TP, and shows that the dust number has smaller contribution for total aerosols number compared with sulfate, OM and other anthropogenic aerosols in the column."

Generally, the people pay more attention to research the impact of dust on clouds, precipitation, and diabatic heating (e.g., Liu et al., 2019; Ji et al., 2016; Lau et al., 2018), because it is important to understanding the climate change over the TP. However, it is not clear how dust aerosols may affect those critical factors in the climate system over the TP. Therefore, in future studies, we will study the impact of dust on clouds, precipitation, and diabatic heating. Also, we have added the discussion on page 22 line 608-610 as: "However, the impact of dust aerosols on cloud properties and precipitation, and the diabatic heating of dust over the TP are not discussed, which will be investigated in future studies." In addition, both texts and figures are revised as the reviewer suggested.

DeMott, P. J., Prenni, A., Liu, X., Kreidenweis, S. M., Petters, M. D., Twohy, C. H., Richardson, M. S., Eidhammer, T., and Rogers, D. C.: Predicting global atmospheric ice nuclei distributions and their impacts on climate, Proc. Natl. Acad. Sci. USA, 107, 11217–11222, doi:10.1073/pnas.0910818107, 2010.

Huang, J., Wang, T., Wang, W., Li, Z., and Yan, H.: Climate effects of dust aerosols over East Asian arid and semiarid regions, J. Geophys. Res.-Atmos., 119, 11398–11416, doi:10.1002/2014JD021796, 2014.

Ji, Z., Kang, S., Zhang, Q., Cong, Z., Chen, P., and Sillanpää, M.: Investigation of mineral aerosols radiative effects over High Mountain Asia in 1990–2009 using a regional

climate model, Atmos. Res., 178–179, 484-496, 2016.

Lau, K. K., and Kim, K. M.: Impact of Snow Darkening by Deposition of Light-Absorbing Aerosols on Snow Cover in the Himalayas–Tibetan Plateau and Influence on the Asian Summer Monsoon: A Possible Mechanism for the Blanford Hypothesis, Atmosphere, 9, 438; doi:10.3390/atmos9110438, 2018.

Li, Z., Lau, W. K., Ramanathan, V., Wu, G., Ding, Y., Manoj, M. G., Liu, J., Qian, Y., Li, J., Zhou, T., Fan., J., Rosenfeld, D., Ming, Y., Wang, Y., Huang, J., Wang, B., Xu, X., Lee, S.-S., Cribb, M., Zhang, F., Yang, X., Zhao, C., Takemura, T., Wang, K., Xia, X., Yin, Y., Zhang, H., Guo, J., Zhao, P., Sugimoto, N., Babu, S. S., and Brasseur, G. P.: Aerosol and monsoon climate interactions over Asia, Rev. Geophys., 54, 866–929, 2016.

Liu, Y., Zhu, Q., Huang, J., Hua, S., and Jia, R., Impact of dust-polluted convective clouds over the Tibetan Plateau on downstream precipitation, Atmos. Environ., 209, 67-77, 2019.

Xu, B., Cao, J., Hansen, J., Yao, T., Joswiak, D. R., Wang, N.,Wu, G.,Wang, M., Zhao, H., Yang,W., Liu, X., and He, J.: Black soot and the survival of Tibetan glaciers, P. Natl. Acad. Sci. USA, 106, 22114–22118, 2009.

Yao, T., Thompson, L., Yang, W., Yu, W., Gao, Y., Guo, X., Yang, X., Duan, K., Zhao, H., Xu, B., Pu, J., Lu, A., Xiang, Y., Kattel, D. B., and Joswiak, D.: Different glacier status with atmospheric circulations in Tibetan Plateau and surroundings, Nat. Clim. Change, 2, 663–667, doi:10.1038/nclimate1580, 2012.

Zhao, C., Chen, S., Leung, L. R., Qian, Y., Kok, J. F., Zaveri, R. A., and Huang, J.: Uncertainty in modeling dust mass balance and radiative forcing from size parameterization, Atmos. Chem. Phys., 13, 10733–10753, doi:10.5194/acp-13-10733-2013, 2013.

*Specific comments:*

- *1. The references in the text should not be limited in the aerosol community, especially for the climate effect of the TP in terms of dynamical and thermal effects.*

**Response:** Thank you so much for your suggestions. We have added discussion about the importance of the TP in modulating the regional climate in page 3-4 line 74-88 as: "The TP exerts profound influences on the climate system in the Northern Hemisphere by modulating the thermodynamics of the atmosphere (Yeh et al., 1957; Ye and Wu et al., 1998; Wu et al., 2007). Generally, the TP serves as a huge heat source in mid- and uppertroposphere that largely affects the Asian summer monsoon in summer, while in winter the TP as a giant wall across the Eurasian continent can block cold outbreaks from the north and confines the winter monsoon to the eastern and southern Asia (Wu et al., 2012). Recent studies have indicated that the climate of the TP is warming rapidly (e.g., Duan and Wu et al., 2008; Wang et al., 2008; Xu et al., 2009; Kang et al., 2010) and the glacier retreat is accelerating (Yao et al., 2007). These changes over the TP are primarily attributed to increasing greenhouse gases (e.g., Duan et al., 2006; Ren et al., 2006), but other major factors could also contribute to these changes, such as absorbing aerosols (e.g., dust and black carbon), which can heat atmosphere and reduce the snow albedo (Kang et al., 2000; Lau et al., 2006, 2010; Xu et al., 2009). The intercontinental transport of dust aerosols originated from surrounding major deserts, i.e., Gobi and Taklamakan deserts on the northern side and Sahara and Arabian deserts on the western side, frequently attacks the TP (Huang et al., 2008; Liu et al., 2008) and greatly influences the regional climate over the TP (Lau et al., 2006; Wu et al., 2007)."

Duan, A. M. and Wu, G. X.: Weakening trend in the atmospheric heat source over the Tibetan Plateau during recent decades. Part I: Observations, J. Clim., 21, 3149–3164, doi:10.1175/2007JCLI1912.1, 2008.

Duan, A., Wu, G., Zhang, Q., and Liu, Y.: New proofs of the recent climate warming over the Tibetan Plateau as a result of the increasing greenhouse gases emissions, Chin. Sci. Bull., 51, 1396–1400, doi:10.1007/s11434-006-1396-6, 2006.

Huang, J., Minnis, P., Chen, B., Huang, Z., Liu, Z., Zhao, Q., Yi, Y., and Ayers, J.: Long-range transport and vertical structure of Asian dust from CALIPSO and surface measurements during PACDEX, J. Geophys. Res., 113, D23212, doi:10.1029/2008JD010620, 2008.

Lau, K. M., Kim, M. K., and Kim, K. M.: Asian monsoon anomalies induced by aerosol direct forcing: the role of the Tibetan Plateau, Clim. Dyn., 26, 855–664, 2006.

Lau, K. M., Kim, M. K., Kim, K.-M., and Lee, W. S.: Enhanced surface warming and accelerated snow melt in the Himalayas and Tibetan Plateau induced by absorbing aerosols, Environ. Res. Lett., 5, 025204, doi:10.1088/1748-9326/5/2/025204, 2010.

Liu, Z., Omar, A., Vaughan, M., Vaughan, M., Hair, J., Kittaka, C., Hu, Y., Powell, K., Trepte, C., Winker, D., Hostetler, C., Ferrare, R., and Pierce, R.: CALIPSO lidar observations of the optical properties of Saharan dust: A case study of long-range transport, J. Geophys. Res., 113(D7):D07207, 2008.

Kang, S.,Wake, C., Qin, D., Mayewski, P. A., and Yao, T.: Monsoon and dust signals recorded in Dasuopu glacier, Tibetan Plateau, J. Glaciol., 46, 222–226, 2000.

Kang, S., Wei, X., You, Q., Flugel, W., Pepin, N., and Yao, T.: Review of climate and cryospheric change in the Tibetan Plateau, Environ. Res. Lett., 5, 015101, doi:10.1088/1748-9326/5/1/015101, 2010.

Ren, J., Jing, Z., Pu, J., and Qin, X.: Glaciers variations and climate change in the central Himalaya over the past few decades, Ann. Glaciol., 43, 218–222, 2006.

Wang, B., Bao, Q., Hoskins, B.,Wu, G., and Liu, Y.: Tibetan Plateau warming and precipitation changes in East Asia, Geophys. Res. Lett., 35, L14702, doi:10.1029/2008GL034330, 2008.

Wu, G., Liu, Y., Wang, T., Wan, R., Liu, X., Li, W., Wang, Z., Zhang, Q., Duan, A., and Liang, X., The Influence of Mechanical and Thermal Forcing by the Tibetan Plateau on Asian Climate, J. Hydrometeorol., 8, 770–789, 2007.

Wu, G., Liu, Y., He, B., Bao, Q., Duan, A., and Jin, F.-F.: Thermal controls on the Asian summer monsoon, Sci. Rep., 2, 404, doi:10.1038/srep00404, 2012.

Xu, B., Cao, J., Hansen, J., Yao, T., Joswiak, D. R., Wang, N.,Wu, G.,Wang, M., Zhao, H., Yang,W., Liu, X., and He, J.: Black soot and the survival of Tibetan glaciers, P. Natl. Acad. Sci. USA, 106, 22114–22118, 2009.

Yao, T., Pu, J., Lu, A., Wang, Y., and Yu, W.: Recent glacial retreat and its impact on hydrological processes on the Tibetan Plateau, China, and surrounding regions, Arct. Antarct. Alp. Res., 39, 642–650, 2007.

Ye, D. andWu, G.: The role of the heat source of the Tibetan Plateau in the general circulation, Meteorol. Atmos. Phys., 67, 181–198, doi:10.1007/BF01277509, 1998.

Yeh, T., Lo, S., and Chu, P.: The wind structure and heat balance in the lower troposphere over Tibetan Plateau and its surrounding, Acta Meteor. Sinica, 28, 108–121, 1957.

- *2. Abstract. "The East Asian dust transports southward and is lifted up to the TP". This statement seems to be misleading because East Asia area defined here, i.e., (25º N – 50º N and 75º E – 150º E) covers a large domain with the TP itself included. According the context, it might mainly be Gobi and Taklamakan. A more specific description is needed.*

**Response:** Thank you so much for your suggestions. In the updated model, we divide the dust sources into four source regions: East Asia (25° N – 50° N and 75° E – 150° E), North Africa (0° N – 40° N and 20° W – 35° E), North America (15° N – 50° N and 80° W – 140°

W), and the elsewhere in the world (dominated by the Middle East in the Northern Hemisphere, 0 – 50° N and 35° E – 75° E). Over the East Asian source region, the dust emission mainly appears in the Gobi and Taklamakan deserts, while the dust emission over the TP is much smaller. So, we think that the long-range transport East Asia dust is mainly from the Gobi and Taklamakan deserts. In order to better showing the results, we have changed the sentence in abstract to "The East Asian dust (mainly from Gobi and Taklamakan deserts) transports southward and is lifted up to the TP," in page 2 line 32, and a specific description as "Clearly, dust emission from the TP is much smaller than that of the Gobi and Taklamakan deserts, so the East Asian dust defined here is mainly contributed by that emitted from Gobi and Taklamakan deserts." in page 8 line 198-199.

- *3. Introduction. "As the highest (about 4000 m) and largest plateau in the world, the Tibetan Plateau (TP): : :", the largest plateau is not true.*

**Response:** Thank you so much for your suggestions. We have changed it to "As the highest plateau (averaged elevation about 4000 m) in the world, the Tibetan Plateau (TP) has received extensive scientific attentions in the past decades due to its unique topographic characteristics (Ma et al., 2014; Yao et al., 2012)."

- *4. Introduction. The conception of "elevated heat pump" can be traced to some earlier literatures.*

**Response:** Thank you so much for your suggestions. We have been traced to some earlier literatures (e.g., Lau et al., 2006; Lau and Kim, 2006; Nigam and Bollasina, 2010; D'Errico et al., 2015).Also, In this study, we have changed this sentence to "Because of the strong ability to absorb and scatter solar radiation, dust can heat up the air in the mid-to-upper troposphere (Lau et al., 2006; Lau and Kim, 2006; Nigam and Bollasina, 2010; D'Errico et al., 2015), which provides cyclonic conditions in the lower layer and induces dry convention over the TP (Ma et al., 2011; Xu et al., 2018)." in page 4 line 98-102.

D'Errico, M., Cagnazzo, C., Fogli, P. G., Lau, K. M., von Hardenberg, J., Fierli, F. and Cherchi, A.: Indian monsoon and the elevated-heat-pump mechanism in a coupled aerosol-climate model, J. Geophys. Res. Atmos., 120, 8712–8723, doi:10.1002/2015JD023346, 2015.

Lau, K. M., Kim, M. K., and Kim, K. M.: Asian summer monsoon anomalies induced by aerosol direct forcing: the role of the Tibetan Plateau, Clim. Dyn., 26, 855–864, 2006.

Lau, K. M. and Kim, M. K.: Observational relationships between aerosol and Asian monsoon rainfall, and circulation, Geophys. Res. Lett., 33, L21810, doi:10.1029/2006GL027546, 2006.

Ma, Y., Zhong, L., Wang, B., Ma, W., Chen, X., and Li, M.: Determination of land surface heat fluxes over heterogeneous landscape of the Tibetan Plateau by using the MODIS and in situ data, Atmos. Chem. Phys., 11, 10461–10469, https://doi.org/10.5194/acp-11-10461-2011, 2011.

Nigam, S., and Bollasina, M.: "Elevated heat pump" hypothesis for the aerosol-monsoon hydroclimate link: "Grounded" in observations?, J. Geophys. Res., 115, D16201, doi:10.1029/2009JD013800, 2010.

Xu, C., Ma, Y., Yang, K., and You, C.: Tibetan Plateau Impacts on Global Dust Transport in the Upper Troposphere, J. Climate, 31, 4745–4756, 2018.

- *5. Caption of Fig. 1. It should be climate mean spatial distribution.*

**Response:** Thank you so much for your suggestions. We have changed the caption to "**Figure 1.** Spatial distribution of annual mean (a) dust emission ($\mu g/m^2/s$) and (b) elevation (m) simulated in WRF-Chem for the period of 2010–2015." in page 33 line 966-967.

- *6. Section 4. Evaluation of AOD and DOD simulated by WRF-Chem. "Between 6–9 km and 9–12 km, the modeled DOD are higher than CALIPSO. The reason would be that the CALIPSO nighttime retrievals in cloud-free condition are used, which have higher accuracy than daytime observations (Winker et al., 2009). Besides, the model results are averaged in all day." This explanation for model bias seems to be questionable, is it possible related to circulation condition? How about the nighttime results for model? Also, it's better to put CALIPSO results in the left panels of Fig. 3.*

**Response:** Thank you so much for your suggestions. We compare the model nighttime results against the CALIPSO nighttime retrievals. It shows that evident differences still exist between the model results and the CALIPSO retrievals. Therefore, we change the sentence to "Between 6–9 km and 9–12 km, the modeled DOD are higher than CALIPSO, which is possibly induced by model biases, such as those related to circulation condition."

at page 12 line 312-314. Also, we have changed the Figure 3 (below).

[Figure]

**Figure 3.** Spatial distributions of annual mean DOD from CALIPSO (left) and WRF-Chem (right) between 0–3 km, 3–6 km, 6–9 km, and 9–12 km for the period of 2010–2015.

- *7. The TP domain plotted in Fig. 4 is too large, with many surrounding plains included such as north Indian subcontinent and north Indo-China Peninsula.*

**Response:** Thank you so much for your suggestions. We have changed TP domain plotted in Figure 4. In order to better describe the dust transport over the TP, we extend the south fringe of the TP at 23$^o$ N in Figure 4, which is similar with that in Xu et al. (2018).

[Figure]

**Figure 4a.** Spatial distributions of (Left) annual mean dust mass loading (mg/m²) emitted from East Asia between 0–3 km, 3–6 km, 6–9 km, and 9–12 km and (Right) the corresponding fractions compared with total dust mass loading over TP. The box in red dotted lines shows the dust source regions, while the box in blue solid lines shows the TP regions in this study. The purple solid lines are at 75° E, 90° E and 102° E for the dust height–latitude cross sections.

[Figure]

**Figure 4b.** Same as figure 4a, but for dust emitted from North Africa. The purple solid lines are at 24° N, 36° N and 38° N for the dust height–longitude cross sections.

[Figure]

**Figure 4c.** Same as figure 4a, but for dust emitted from the Middle East.

- *8. Fig. 5. The annual mean horizontal and vertical wind field may neglect the remarkable seasonality of TP and adjacent monsoon regions. Particularly for the ascending at 3-6 km but descending above it in the south fringe of the TP.*

**Response:** Thank you so much for your suggestions. We have added the seasonal mean horizontal and vertical wind field in figure S3. The corresponding discussion is shown in page 14 line 373-378 as "Also, the horizontal and vertical wind has significant seasonal variability (Fig. S3). In the 0-3 km, the maximum updrafts appear in JJA over major dust source regions, followed by MAM. Over the TP, the updrafts are clearly in JJA, while the downdrafts are dominated in DJF. In MAM and SON, the updrafts and downdrafts occur in a spatially uniform way. Additionally, the vertical wind is ascending at 3-6 km in the south fringe of the TP across all seasons, while the vertical wind is descending above 6 km in the DJF, MAM and SON."

[Figure]

**Figure S3.** Seasonal mean wind field between 0–3 km, 3–6 km, 6–9 km, and 9–12 km from WRF-Chem simulations for the period of 2010–2015. The arrows indicate the horizontal wind (m/s). The colors indicate the vertical wind velocity (m/s) and positive values are for updrafts and negative values are for downdrafts.

- ***9. Section 5.4 Dust radiative forcing over the TP. "the dust LW radiative forcing at TOA is waring with a positive value of 0.5 W/m2". Waring is a typo.***

**Response:** Thank you so much for your suggestions. We have change it to "warming".

- ***10. Fig. 9. The average altitude of TP is 4 Km, how can get the results at 0-3 Km?***

**Response:** Thank you so much for your question. We have add the description in page 15-16 line 432-434 as: "It should be noted that the TP is defined as 27º N – 38º N and 75º S – 103º S (Fig. S4), which shows that some areas within the TP domain have altitude of smaller than 3 km." So, we can calculate the dust mass flux at 0-3 km.

[Figure]

**Figure S4.** Spatial distribution of annual elevation (m) simulated in WRF-Chem for the period of 2010–2015.

- ***11. Fig. 10 and 12 can be merged in one.***

**Response:** Thank you so much for your suggestions. We have merged them as Figure 10a and Figure 10b.

- ***12. Section 6 Conclusions and Discussion. "the dust particle can reach upper-troposphere (above 8 km) and even stratosphere, which can provide a pathway for dust into the upper-troposphere (above 9 km)." This expression is strange and hard to follow.***

**Response:** Thank you so much for your suggestions. We have changed it to "Under the effect of TP and strong Indian summer monsoon, the dust particles can reach the upper-troposphere (above 8 km) and even lower-stratosphere. This study proposes a pathway for dust transport into the upper-troposphere, and provides a climatological view about the intercontinental transport characteristics of dust from different source regions over the TP." in page 21-22 line 598-601.

- ***13. Caption of Fig. 13. "Spatial distribution of seasonal mean: : :". For which season the authors mentioned here.***

**Response:** Thank you so much for your suggestions. We have changed it to "**Figure 12.** Spatial distribution of annual mean SW, LW, and NET (SW + LW) direct radiative forcing (W/m$^2$) of dust at the TOA (top panels), in the atmosphere (ATM: middle panels) and at the surface (SFC: bottom panels) under all-sky conditions from the WRF-Chem simulation averaged for 2010–2015."

**Anonymous Referee #3**

*General comments:*

- *This study provides long-term quasi-global simulations of dust aerosols and investigates the dust source contribution and transport over the Tibetan Plateau (TP). By a tracer-tagging technique, the dust emitted from different deserts are identified and the contributions of dusts from the global major deserts to the TP are quantitively estimated. Furthermore, the dust radiative effects over the TP are also studied. The manuscript is well organized and written. Therefore, I recommend publishing it once the following comments are addressed.*

**Response:** We thank the reviewer for a detailed review. Both text and figures are revised as the reviewer suggested.

*Specific comments:*

- *1. The simulation is conducted with the CBMZ-MOSAIC scheme, please briefly introduce how the anthropogenic gas and aerosol emissions form the HTAPv2 are mapped to the CBMZ-MOSAIC species. Otherwise, it is difficult for readers to reproduce the experiment.*

**Response:** Thank you so much for your suggestions. We have add the briefly introduction of HTAPv2 in the Supplementary materials as "The HTAP_V2 dataset consists of $CH_4$, CO, $SO_2$, NOx, NMVOC, $NH_3$, $PM_{10}$, $PM_{2.5}$, BC and OC. The anthropogenic gases are $CH_4$, CO, $SO_2$, NOx, NMVOC and $NH_3$, and the aerosol particles are $PM_{10}$, $PM_{2.5}$, BC and OC. For the anthropogenic gases, the $CH_4$, CO, $SO_2$, NOx and $NH_3$ can be mapped directly to the CBMZ-MOSAIC species. However, the NMVOC emissions are calculated for 23 VOCs species, using the VOCs breakdown that was generated by the TNO (boxlat1). Also, The RETRO ratios are applied to HTAPv2 total NMVOC emissions files for the five HTAPv2 sectors (Power, Industry, Transport, Residential, Agriculture), for which individual VOC can be calculated. More details on the HTAPv2 VOCs breakdown are available online at: http://iek8wikis.iek.fz-juelich.de/HTAPWiki/WP1.1?highlight=%28%28WP1.1%29%29. Moreover, the aerosol particles (include $PM_{10}$, $PM_{2.5}$, BC and OC) can be mapped directly to the MOSAIC species."

- *2. Could you briefly explain the tracer-tagging technique used in the study without reader to refer to other reference.*

**Response:** Thank you so much for your suggestions. We have briefly explained the tracer-tagging in page 8-9 line 219-227 as: "The dust particles emitted from the four source regions are tagged and explicitly tracked using additional model variables within a single simulation in the model following Hu et al. (2016, 2019a, 2019b) and Mao et al. (2019). During the model simulation, the physical and advective tendencies of all tagged dust variable are treated in the same way. Therefore, the contribution of a dust source to the dust property in a receptor region is defined by the ratio of the dust property in the receptor region from the dust source to the dust property of all sources combined. This direct tagging technique neither introduces aerosol source perturbations to the model nor employs assumptions for aerosol sources/sinks along the transport pathways (Hu et al., 2019a; Mao et al., 2019)."

Hu, Z., Huang, J., Zhao, C., Bi, J., Jin, Q., Qian, Y., Leung, L. R., Feng, T., Chen, S., Ma, J.: Modeling the contributions of Northern Hemisphere dust sources to dust outflow from East Asia, Atmos. Environ., 202, 234–243, 2019a.

Hu, Z., Huang, J., Zhao, C., Ma, Y., Jin, Q., Qian, Y., Leung, L. R., Bi, J., and Ma, J.: Trans-Pacific transport and evolution of aerosols: spatiotemporal characteristics and source contributions, Atmos. Chem. Phys., 19, 12709–12730, https://doi.org/10.5194/acp-19-12709-2019, 2019b.

Mao, R., Z. Hu, C. Zhao, D. Gong, D. Guo, G. Wu: The source contributions to the dust over the Tibetan Plateau: A modelling analysis, in review, Atmos. Environ., 2019.

- *3. How the dust radiative forcing is calculated in this study? Is it the double call of the radiative module or other method?*

**Response:** Thank you so much for your suggestions. In this study, the dust radiative forcing is calculated as double call of the radiative module following Zhao et al. (2013). More specifically, the calculation of aerosol radiative transfer is performed multiple times with the mass of one or more aerosol species (i.e., the mass of an individual or a group of aerosol species). After this diagnostic iteration procedure, the aerosol direct radiative forcing for an individual or a group of species are estimated by subtracting the optical

properties and direct radiative forcing in the diagnostic iterations from those in the standard procedure including all the aerosol species (Zhao et al., 2013). So, the dust radiative forcing can be calculated as:

$$\text{Forcing}_{[dust]} = \text{Forcing}_{[all\text{-}aerosol]} - \text{Forcing}_{[without\ dust]}$$

[revised manuscript text omitted]